# A Systematic Review on the Research Progress on Polysaccharides from Fungal Traditional Chinese Medicine

**DOI:** 10.3390/molecules28196816

**Published:** 2023-09-27

**Authors:** Chenxi Bai, Fazhi Su, Wensen Zhang, Haixue Kuang

**Affiliations:** Key Laboratory of Basic and Application Research of Beiyao, Ministry of Education, Heilongjiang University of Chinese Medicine, Harbin 150040, China; bcx19990709@163.com (C.B.); sfz18406564303@163.com (F.S.); zhang1033362077@163.com (W.Z.)

**Keywords:** fungal traditional Chinese medicine, polysaccharide, structural identification, biological functions, quality control, application

## Abstract

Traditional Chinese medicine (TCM) is a class of natural drugs with multiple components and significant therapeutic effects through multiple targets. It also originates from a wide range of sources containing plants, animals and minerals, and among them, plant-based Chinese medicine also includes fungi. Fungal traditional Chinese medicine is a medicinal resource with a long history and widespread application in China. Accumulating evidence confirms that polysaccharide is the main pharmacodynamic material on which fungal TCM is based. The purpose of the current systematic review is to summarize the extraction, isolation, structural identification, biological functions, quality control and medicinal and edible applications of polysaccharides from fungal TCM in the past three years. This paper will supplement and deepen the understanding and application of polysaccharides from fungal TCM, and propose some valuable insights for further research and development of drugs and functional foods.

## 1. Introduction

Fungi are a group of eukaryotic organisms with a diverse variety (including microorganisms and the more familiar mushrooms). They have benefits that play a crucial role in balancing the environment and generating economic benefits [1]. The ways of administering fungi can be mainly divided into two categories: (1) they can be directly administered or co-administered with other herbal medicines to treat various diseases; and (2) they can be produced as viable foods in a variety of ways to achieve health care. For example, Auricularia auricular-judae is a kind of fungus that is homologous to medicine and food, and its biological function was recorded as early as in Shen Nong’s Herbal Classic. It is reported that dietary intake of *Auricularia auricular-judae* can alleviate hypertension, vascular sclerosis and gastric disorders [2]. In modern medicinal research, multiple pharmacological functions have been demonstrated, for example, anti-coagulant, anti-neoplastic, hypoglycemic, hypocholesterolemic and anti-platelet aggregation [3,4,5,6]. This also suggests that the full development and utilization of fungi in the pharmaceutical and functional health food industries has broad prospects.

Polysaccharide, a kind of biological macromolecule formed by dehydration and condensation of more than 10 monosaccharide molecules, possesses a series of biological activities [7,8,9]. For the past few years, an increasing amount of polysaccharides derived from herbal medicine were purified and structurally identified [10,11,12,13,14,15]. At the same time, the various biological functions of polysaccharides from Chinese medicine containing plants, animals and microorganisms have also attracted increasing attention [16,17]. For example, the anti-inflammatory, immunomodulatory, antioxidant, hepatoprotective, hypolipidemic, hypoglycemic and gut microbiota regulatory activities of polysaccharides as well as their potential mechanisms have been elucidated [18,19,20,21,22]. Additionally, deciphering their structure–activity relationships help us to understand polysaccharides to a greater degree.

Through literature review, it was found that polysaccharide is an important active ingredient of fungal traditional Chinese medicine, such as *Ganoderma lucidum*, *Auricularia auricular-judae* and *Portulaca oleracea*, etc. [23,24,25]. In terms of the research progress of *Ganoderma lucidum* polysaccharide, the chemical structure, biological activity and corresponding structure–activity relationship have been deciphered [26,27]. Furthermore, extensive investigation has been conducted on multiple species of *Ganoderma lucidum*. As another example, researchers have recently isolated and purified a novel and unprecedented glucuronoxylogalactoglucomannan from *Auricularia auricular-judae*, and preliminarily evaluations have confirmed its anti-inflammatory activity [28]. This also inspires us to continue to explore the novel structure and broad biological activities of carbohydrates from fungal traditional Chinese medicine urgently, due to their rich medicinal and edible value.

Interestingly, it was found that there is still a lack of comprehensive understanding of polysaccharides in fungal traditional Chinese medicine polysaccharides based on the review of the existing literature. Against this background, polysaccharides from fungal traditional Chinese medicine (included in the Chinese Pharmacopoeia, 2020 edition) including *Ganoderma lucidum*, *Cordyceps sinensis*, *Poria cocos*, *Polyporus*, *Omphalia* and *Lasiosphaera calvatia* were selected as the focus of investigation in conducting our literature review. We systematically reviewed the research status of these medicines from the perspective of preparation methods, structural identification, chemical properties and biological activities in the current article. We hope that the summary can provide a strong basis for the further development and application of fungal polysaccharides, especially in exploring their medicinal and edible value.

## 2. Preparation

With increasing attention being paid to the medicinal and edible value of fungal TCM, a series of conventional and modern extraction means have been developed and applied [29]. Interestingly, it has been documented that fungal polysaccharides are classified into intracellular polysaccharides and exopolysaccharides based on their distribution and cellular localization [30]. In general, raw materials should be subjected to a series of pretreatments prior to extraction of intracellular polysaccharides. Hot water extraction is the most common choice to extract polysaccharides from fungal TCM because of its easy operation, low economic cost and convenient use [31]. Additionally, dilute alkali method, a traditional technology, is also established and applied to extract the polysaccharides from fungal TCM [32]. Unfortunately, the traditional solvent extraction methods also exhibit some disadvantages including high temperature, long extraction duration and unnecessary structural failure [33]. To address these limitations, some novel effective extraction means have been explored in accordance with the aim of promoting cell wall breakdown without destroying the structure of polysaccharides from fungal TCM. The specific extraction methods and corresponding parameters are listed in Table 1. Meanwhile, the preparation process of fungal TCM polysaccharides is provided in Figure 1.

### 2.1. Extraction of Polysaccharides from Fungal TCM

Liu et al. [34] used a continuous phase transition extraction apparatus to extract crude polysaccharides from the fruiting bodies of *Ganoderma lucidum*. In a study from Vietnam, red *Ganoderma lucidum* was selected for the exploration of an optimal extraction process for its polysaccharides. The response surface methodology and Box–Behnken design were adopted to explore the effects of pH, extraction temperature, time and ultrasonic power on the content of polysaccharides [35]. The optimal extraction parameters were as follows: enzyme concentration of 3%, pH 5.5, temperature of 45 °C, time of 30 min and ultrasonic power of 480 W.

Internal boiling method is a method that uses a low-boiling point solvent to infiltrate the material, and then adds the extraction solution with a temperature higher than the boiling point of the infiltrating solvent. Thus, the low-boiling point solvent inside the material causes internal boiling and convective diffusion inside and outside the cell, so as to quickly extract the active ingredient. Tian et al. conducted the internal boiling method to optimize the extraction parameters of *Cordyceps sinensis* polysaccharide, which provided an experimental support to vigorously develop the *Cordyceps sinensis* polysaccharide [36]. Zha et al. extracted the polysaccharide from *Ophiocordyceps sinensis mycelium* using microwave extraction method under the following extraction conditions: mesh number of 80, solid–liquid ratio of 1: 26, extraction frequency of three, microwave power of 330 W and microwave time of 4 min [37]. Mayuree et al. [38] found that when the water extraction time was 24 h under ambient temperature, the extractive yield of polysaccharides from *Cordyceps militaris* of 4.95% can be reached. Wang et al. [39] obtained a new monomer polysaccharide from *Cordyceps militaris* using an alkali extraction method. After the fruiting bodies were defatted, the dried residues were extracted with distilled water, and subsequently, the remaining residues were further extracted with 0.5 mol/L NaOH solution to obtain the alkali-extracted products. 

Polysaccharide is the main bioactive substance in *Poria*, which mainly exists in the cell wall. Some researchers adopted steam blasting technology to pretreat *Poria* and achieved polysaccharide extraction via water extraction and alcohol precipitation method [40]. The main principle of steam blasting technology is to subject the material to a high temperature and high pressure environment, soften the material through water vapor penetration, and expand the gas in the cell rapidly when the pressure is released instantly, so as to achieve the effect of blasting. Subsequently, the separation of the material components is realized through the destruction of the cell wall structure of the material, and this achieves the purpose of increasing the dissolution rate of components [41]. Hu extracted polysaccharide from *Poria cocos* (Schw.) Wolf. using dilute alkali leaching, and results showed that the method increased the polysaccharide content compared with water extraction and alcohol precipitation [42]. Xiang et al. utilized combined enzyme extraction method to optimize the polysaccharide extraction from *Poria cocos*, significantly enhancing the polysaccharide yield compared with water extraction [43]. Huang et al. [44] established a D-optimal mixture design method combined with ultrasonic-assisted enzyme extraction on the basis of single factor experiment to optimize the composition of composite enzymes and the extraction process parameters. The method was stable and feasible, and increased the extraction efficiency of *Poria* polysaccharide, which can provide a reference for the development and utilization of *Poria* resources. 

*Polyporus* polysaccharide is a kind of polar macromolecule compound with a complex structure, and it usually exists in the cell wall. Hence, optimal extraction method should be selected to obtain the *Polyporus* polysaccharide. Zheng et al. [45] applied ultrasonic-assisted extraction method to extract polysaccharides from *Polyporus*. Meanwhile, they improved extraction efficiency through response surface methodology, and the specific extraction parameters were as follows: extraction temperature 72 °C, extraction power 300 W, extraction time 65 min and liquid-to-solid ratio 22 mL/g. However, when using microwave-assisted extraction [46] under the following optimal extraction conditions, the yield of polysaccharides could reach 6.75%: liquid-to-solid ratio of 30:1, pH 6.6, extraction time of 2.5 min and microwave power of 614 W. Bi Yunpeng et al. pretreated *Polyporus umbellatus* with 80% ethanol for 24 h before polysaccharide extraction to remove the interfering components. Then, the residue was extracted with hot distilled water to obtain the polysaccharide [47]. In order to obtain a greater variety of polysaccharides, different concentrations of dilute alkali–water solutions were usually adopted for extracting the remaining polysaccharides from the residue obtained after hot water extraction. For example, Ueno et al. [48] obtained several polysaccharides from *Polyporus* using alkali solution extraction method with 2% sodium hydroxide containing 2% of urea in the 1980s. In addition, considering the extraction efficiency, cost and other factors, enzymolysis combined with ultrasonic-assisted extraction method has also been reported to extract polysaccharides from *Polyporus*. Zhang et al. [49] optimized the extraction process of *Polyporus* polysaccharide with the orthogonal test. In the study, the yield of the *Polyporus* polysaccharide was up to 4.31% under the following condition of cellulose/pectinase = 1:1, temperature = 50 °C, and pH = 6.5 for 60 min. Surprisingly, after an additional 30 min of ultrasonic treatment, the yield increased to 7.48%. 

Bi et al. [50] optimized the extraction condition of *Omphalia* polysaccharides based on response surface method. The study finally determined the optimal conditions of polysaccharides extraction to be as follows: the water/solid ratio of 21:1, extraction temperature of 100 °C, the extraction time of 93 min and an extraction yield of up to 23.12% with twice extraction. Xu et al. [51] applied cellulose method to investigate the optimal extraction process of *Omphalia* polysaccharides. The results showed that the best extraction condition was as follows: enzymolysis temperature of 40 °C, enzyme addition of 0.08%, enzymolysis time of 90 min and a final extraction rate of polysaccharides of up to 3.57%. 

As an important active component, the polysaccharide contained in *Lasiosphaera calvatia* has been paid increasing attention in recent years because of its functions of protecting the liver and scavenging superoxide free radicals [52,53]. Shi et al. [54] extracted polysaccharides from *Lasiosphaera fenzlii* using ultrasound-assisted extraction. Furthermore, they optimized the extraction process parameters based on the central composite design response surface method that are as follows: liquid-to-solid ratio of 30:1; reflux time of 53.77 min and extraction temperature of 58.33 °C. The snailase-assisted extraction with the advantage of high yield, short extraction time and mild condition was also used to extract polysaccharide from *Lasiosphaera* [55]. It was found that the yield of polysaccharide was the highest under the following conditions: extraction temperature of 35 °C; extraction time of 120 min; snailase addition of 5.0% and pH 6.0. Yang et al. [56] found that the wall-breaking process could improve the extraction efficiency of polysaccharides from *Lasiosphaera*. They adopted high-temperature water extraction at 100 °C for 2.5 h and alcohol precipitation combined with wall-breaking method to extract polysaccharides. 

**Table 1 molecules-28-06816-t001:** Extraction methods and parameters of polysaccharides extraction from fungal TCM.

Fungi Name	Polysaccharide Extraction Method	Parameters	Total Yield (%)	Reference
*Ganoderma lucidum*	Continuous phase transition extraction	The air-dried fruiting bodies of *Ganoderma lucidum* were extracted according to the following conditions: distilled water, 100 °C, 4 h, and flow rate of 28 L/h	7.13%	[34]
*Red Ganoderma lucidum*	Ultrasound-assisted enzymatic extraction	Enzyme concentration, 3%; pH, 5.5; extraction temperature, 45 °C; extraction time, 30 min; ultrasonic power, 480 W	The highest content of polysaccharides was 32.08 mg/g	[35]
*Cordeceps sinensis*	Internal boiling method	Resolver concentration, 90%; parse time, 1 min; extractant volume, 60 mL; extract time, 4 min; 100 °C	2.37%	[36]
*Ophiocordyceps sinensis*	Microwave extraction	Mesh number, 80; 1:26 (g/mL); 3 times; microwave power, 330 W; microwave time, 4 min	9.06%	[37]
*Cordyceps militaris*	Water extraction	Shaking (150 rpm); 1: 10 (*w*/*v*); 24 h; twice	4.95%	[38]
*Cordyceps militaris*	Alkali extraction	Extraction solvent, 0.5 M NaOH solution; 8 volumes; 95 °C; 3 h; twice	N/A	[39]
*Poria*	Steam blasting pretreatment, water extraction and alcohol precipitation	Steam blasting pressure, 2.0 MPa; dwell time, 60 s; 1:50 (g/mL); 60 °C; 120 min	1.95%	[40]
*Poria cocos* (Schw.) Wolf	Dilute alkali leaching	4 °C; 0.15 M NaOH solution	72.656% (polysaccharide content)	[42]
*Poria cocos*	Combined enzyme extraction method	Cellulase, 2.5%; hemicellulose, 2.5%; beta-glucanase, 5%; 90 min; 50 °C; pH 5.0	6.13%	[43]
*Poria cocos*	Ultrasonic-assisted enzymatic extraction	Combined enzyme addition amount (cellulase:papain = 4:1), 7.60%; 1:44 (g/mL); pH 4.90	10.42%	[44]
*Polyporus umbellatus*	Ultrasonic-assisted extraction	Extraction temperature, 72 °C; extraction power, 300 W; extraction time, 65 min; liquid-to-solid ratio, 22 mL/g	2.47%	[45]
*Polyporus umbellatus*	Microwave-assisted extraction	Extraction time, 2.5 min; microwave power, 614 W; pH 6.6; liquid-to-solid ratio, 30:1	6.75%	[46]
*Polyporus umbellatus*	Hot distilled water extraction	Pretreated with 80% ethanol for 24 h; hot distilled water, 90 °C; twice; 4 h	N/A	[47]
*Polyporus umbellatus*	Alkali solution extraction	2% sodium hydroxide containing 2% of urea	N/A	[48]
*Polyporus umbellatus*	Enzymatic coupled with ultrasonic-assisted extraction	Cellulose:pectinase = 1:1; 50 °C; pH = 6.5, 60 min; additional 30 min of ultrasonic treatment	7.48%	[49]
*Omphalia*	Water extraction	Extraction temperature, 100 °C; extraction time, 93 min; the water/solid ratio, 21:1; twice	23.12%	[50]
*Omphalia*	Cellulose enzymolysis	Enzymolysis temperature of 40 °C; cellulose addition, 0.08%; enzymolysis time, 90 min	3.57%	[51]
*Lasiosphaera fenzlii*	Ultrasound-assisted extraction	Liquid-to-solid ratio, 30:1; reflux time, 53.77 min; extraction temperature, 58.33 °C	4.01%	[54]
*Lasiosphaera puffball*	Snailase-assisted extraction	Extraction temperature, 35 °C; extraction time, 120 min; snailase addition, 5.0%; pH 6.0	0.908%	[55]
*Lasiosphaera puffball*	High-temperature water extraction and alcohol precipitation as well as wall-breaking method	Breaking speed, 600 rpm; extraction time, 2.5 h; extraction temperature, 100 °C; liquid-to-solid ratio, 20:1	1.065%	[56]

### 2.2. Purification and Isolation of Polysaccharides from Fungal TCM

In order to obtain purer and more homogenous polysaccharide fractions for further structural identification, a series of technical strategies and purification process are conducted to purify crude polysaccharides from fungal TCM [57]. Generally, the crude polysaccharides of fungal TCM are firstly subjected to deproteinization and decolorization. Small molecules mixed in the crude polysaccharides are removed through dialysis method [58]. Afterwards, a series of column chromatography containing ion exchange chromatography and gel filtration chromatography are adopted to carry out further purification with relevant eluents [59]. At last, purified polysaccharide fractions are obtained after concentration, dialysis and lyophilization. The representative flow chart of extraction and purification processes of fungal TCM polysaccharides is presented in Figure 1. The specific information of the purification process for polysaccharides for each fungal TCM is shown in Table 2. 

Chen et al. [60] adopted water extraction and ethanol precipitation to prepare *Ganoderma lucidum* crude polysaccharides. And an acidic β-glucan (GLPC2) was purified through anion-exchange chromatography combined with gel permeation. Milhorini et al. [61] isolated a fucoxylomannan (FXM) from *Ganoderma lucidum* via alkaline extraction followed by dialysis, freeze–thawing, and fractionation with Fehling’s solution. Specifically, the fruit body of *Ganoderma lucidum* was extracted with alkali solution after degreasing, and the alkaline extract was neutralized with acid and dialysis was performed for 72 h. Fehling reagent was added to the solution, and the polysaccharide FXM was obtained after continuous washing with acid and alkali solution and dialysis. Liu et al. [34] performed a continuous phase transition extraction combined with ethanol precipitation method to enrich the polysaccharide fractions from *Ganoderma lucidum*. Subsequently, the crude polysaccharides were injected into the cut-off ultrafiltration membrane to obtain different fractions (EGLP and RGLP) according to their molecular weight. RGLP was further purified through Sephacryl S-500 HR column and eluted with 0.2 mol/L NaCl resolution at the flow rate of 0.8 mL/min. Sheng et al. [62] extracted polysaccharides from *Ganoderma lucidum* using deionized water at 100 °C for 120 min, and ethanol with a final concentration of 60% was used to prepare the crude polysaccharides. The middle-pressure liquid chromatography equipped with a DEAE Sepharose Fast Flow column combined with Sephadex G100 column were used for further purification, and a polysaccharide named GLSP-I was obtained. Dong et al. [63] purified a natural selenized polysaccharide from *Ganoderma lucidum* through water extraction at room temperature. After precipitation with 95% ethanol and deproteinization with Sevag reagent, crude polysaccharide was prepared, and then, DEAE Sepharose Fast Flow column and Sephadex G-100 column were combined to obtain SeMPN. Cai et al. [64] conducted a cascade membrane technology to isolate polysaccharides from *Ganoderma lucidum*. In a nutshell, *Ganoderma lucidum* was extracted with hot water and then graded using 100 kDa, 10 kDa and 1 kDa ultrafiltration membranes in order to achieve transmembrane pressures of 0.6–1 MPa. After deproteinization and precipitation, its further purification was accomplished using DEAE Sepharose fast flow chromatography with different concentrations of NaCl solutions.

In a study from 2021 [39], the researchers purified a novel polysaccharide, CM3-SII, from the water-soluble component of the alkali extract of the *Cordeceps militaris* fruiting body. In this study, two types of column chromatography elution methods were applied, and the water-soluble component was loaded on Sephacryl S200 HR column (2.6 × 90 cm) with 0.2 mol/L NH_4_HCO_3_ as eluent to obtain the monomer polysaccharide. Zhuansun et al. [65] used the traditional DEAE-52 column combined with Sephadex G-100 column to obtain an extracellular homopolysaccharide from fermentation medium of *Cordyceps cicadae*. Furthermore, Tang et al. [66] obtained a novel polysaccharide from *Cordyceps cicadae*, named JCH-a1, which was extracted by ultrasonically assisted enzymatic extraction and purified with cellulose DEAE-32 and Sephadex G-100 column chromatography. Wang et al. [67] extracted, isolated and purified polysaccharides from different parts of *Cordyceps cicadae*. Three polysaccharides were subjected to DEAE-52 column chromatography, collected, concentrated, dialyzed and lyophilized. Similarly, Hu et al. [68] obtained three novel polysaccharides from *Cordyceps militaris* using this method. Sun et al. [69] prepared a novel neutral exopolysaccharide (EPS-III) from culture broth of *Cordyceps militaris*. After the protein was removed using Sevage method and the pigment was removed using AB-8 macroporous resin, the crude polysaccharide was further purified with Sephadex G-200 column chromatography to obtain EPS-III. It is worth noting that in a recent report [70], researchers purified a new acidic exopolysaccharide (AEPS-II) with immunological activity from fermentation broth of *Cordyceps militaris*. In this study, NKA-9 macroporous adsorption resin was used to remove the pigment, and the crude EPS was eluted using DEAE-Sephacel with distilled water and different concentrations of NaCl aqueous solution. Finally, AEPS-II was purified using Sephadex G-200 with distilled water.

*Poria cocos* exhibits therapeutic potential against cancer as a famous traditional Chinese medicine and a well-known food. A glucan was extracted using deep eutectic solvent (ChoCl and oxalic acid in a molar ratio of 1:2) and purified with Sephadex G-15 column chromatography from *Poria cocos*, and its strong antioxidant activity was proved [71]. Lin et al. [72] obtained a mannoglucan from the mycelial culture conditions of *Poria cocos*. The lyophilized mycelia was extracted with a mixture of 0.1 M sodium acetate, 5 mM cysteine, papain and 5 mM EDTA, and then 95% ethanol was added to precipitated crude polysaccharide. The further purification was achieved with dialyzation (cut off at MW 12,000~14,000 Da), centrifugation and lyophilization. Yang et al. [73] obtained a water-insoluble polysaccharide from *Wolfiporia cocos* through alkali extraction and acid precipitation method. In the report of Cheng et al. [74], a galactoglucan (PCP-1C) with hepatoprotective activity was purified from *Poria cocos* sclerotium. After removing the protein and impurities with a molecular weight of less than 3500 Da, the PCP was obtained. Then, the PCP was purified using cellulose DEAE-52 column eluted with NaCl solution. The active fraction was further eluted with ultrapure water on Sephacryl S-500 column to gather PCP-1C. Li et al. [75] prepared four polysaccharides from lyophilized mycelium and fermentation broth of *Poria cocos*, respectively. Traditional DEAE-52 cellulose anion exchange column combined with Sephadex G-100 gel chromatographic column was applied to achieve the purification of these polysaccharides. 

In accordance with the existing literature, in the past three years, glucan was the main polysaccharide in *Polyporus*. As recorded in the research of Liu et al. [76], a new polysaccharide with strong immunomodulatory activity has been isolated with DEAE-52 cellulose column and Sephadex G-100 gel filtration column chromatography from *Polyporus umbellatus*. Gao et al. [77] also isolated a glucan with anti-nonalcoholic steatohepatitis activity from *Polyporus umbellatus*. While different from the above, the crude polysaccharide was extracted by boiling in water in this study, and a DEAE-Sepharose Fast-Flow column was used to separate. The further separation was achieved using Superdex^TM^ G-75 column with 0.2 M NaCl solution. In another report [78], researchers have purified a water-soluble glucan (PGPS) using gel chromatography with Sepharose-6B column from *Polyporus grammocephalus*. 

There are relatively few studies on the separation and purification of polysaccharides from *Omphalia lapidescens*. Rui et al. [79] produced a water-soluble β-glucan from *Omphalia lapidescens* using extraction with 0.5 M NaOH. After washing with water and 0.1 M NaOH, the residue was dissolved in 0.5 M NaOH and acidified with AcOH, and the resulting precipitation was the polysaccharide OL-2.

Coincidentally, there are also fewer studies on the polysaccharides of *Lasiosphaera fenzlii* at present. Xia [80] adopted 0.2 mol/L NaOH solution to extract crude polysaccharide from the residue after alcohol extraction of *Lasiosphaera fenzlii* fruit body. Total polysaccharide was prepared after removal of free protein using trichloroacetic acid and removal of small molecules using dialysis. After preliminarily separation with DEAE cellulose column chromatography and further purification with Sephacryl^TM^ S-200 and Sephacryl^TM^ S-300 gel column chromatography, four homogeneous polysaccharides with anti-complement activity were obtained, named as TFP-1, TFP-2, TFP-3 and TFP-4, respectively. While in an earlier study [81], researchers applied water extraction and ethanol precipitation to obtain crude polysaccharide from *Calvatia geigantea*. The purification was achieved with DEAE-Sepharose fast flow ion-exchange column chromatography and Sephacryl S-300 gel filtration. 

**Table 2 molecules-28-06816-t002:** The purification methods of fungal TCM polysaccharides.

Fungi Name	Source	Polysaccharide Name	Extraction and Purification Method	Reference
*Ganoderma lucidum*	Fruiting body	GLPC2	Water extraction and ethanol precipitation; fractioned using DEAE SepharoseTM FF column with different concentrations of NaCl solution; subsequently eluted using Sephacryl S-200 HR column with 0.5 M NaCl solution	[60]
*Ganoderma lucidum*	Fruiting body	FXM	The fruit body of Ganoderma lucidum was extracted with alkali solution after degreasing, and the alkaline extract was neutralized with acid and dialysis for 72 h. Fehling reagent was added to the solution, and the polysaccharide FXM was obtained after continuous washing with acid and alkali solution and dialysis.	[61]
*Ganoderma lucidum*	Fruiting body	RGLP-1	The crude polysaccharides were injected into the cut-off ultrafiltration membrane to obtain different fractions (EGLP and RGLP) according to their molecular weight. RGLP was further purified through Sephacryl S-500 HR column and eluted with 0.2 mol/L NaCl resolution at the flow rate of 0.8 mL/min	[34]
*Ganoderma lucidum*	Spore	GLSP-I	Water extraction and ethanol precipitation; eluted by different concentrations of NaCl solution using middle-pressure liquid chromatography equipped with a DEAE Sepharose Fast Flow column; then purified on Sephadex G100 column	[62]
*Ganoderma lucidum*	Mycelia	SeMPN	Water extraction and ethanol precipitation; eluted with different concentrations of NaCl solution on a DEAE Sepharose Fast Flow column; then purified on Sephadex G100 column	[63]
*Ganoderma lucidum*	N/A	GLPs	Hot water extraction; graded by ultrafiltration membranes (100 kDa, 10 kDa and 1 kDa); transmembrane pressures of 0.6–1 MPa; flow speed of 700 r/min; deproteinized with Sevag method and precipitated with ethanol; eluted using DEAE Sepharose fast flow chromatography with different concentrations of NaCl solutions	[64]
*Cordyceps militaris*	Fruiting body	CM3-SII	Alkali extraction; water-soluble components were fractionated using Q-Sepharose^TM^ Fast Flow column chromatography with NaCl; CM3-S was then purified on a Sephacryl S200HR column with 0.2 mol/L NH_4_HCO_3_	[39]
*Cordyceps cicadae*	Fermentation medium	PACI-1 (an extracellular selenium-enriched polysaccharide)	PACI solution was loaded onto DEAE-52 column and eluted with pure water and a step gradient of 0.1 M to 0.3 M NaCl solution. Then, the main fraction was eluted with pure water on a Sephadex G-100 column and filtered through 8000 Da molecular mass membranes to desalt	[65]
*Cordyceps cicadae*	Fruiting body	JCH-a1	Ultrasonically-assisted enzymatic extraction (cellulose:chitinase = 1:1); deproteinized with Sevage; the fractions were eluted with DEAE-32 column and Sephadex G-100 column, respectively	[66]
*Cordyceps cicadae*	Bacterium substance	BSP	Hot water bath extraction (78 °C); deproteinized with Sevage; eluted on DEAE-52 column chromatography with different concentrations of NaCl (0, 0.1, 0.2, 0.3, 0.4, 0.5 M NaCl)	[67]
*Cordyceps cicadae*	Spore powder	SPP	Hot water bath extraction (78 °C); deproteinized with Sevage; eluted on DEAE-52 column chromatography with different concentrations of NaCl (0, 0.1, 0.2, 0.3, 0.4, 0.5 M NaCl)	[67]
*Cordyceps cicadae*	Fruiting body	PPP	Hot water bath extraction (78 °C); deproteinized with Sevage; eluted on DEAE-52 column chromatography with different concentrations of NaCl (0, 0.1, 0.2, 0.3, 0.4, 0.5 M NaCl)	[67]
*Cordyceps militaris*	N/A	CMP	Hot water reflux extraction; eluted with 0, 0.1, 0.2, 0.3, 0.4 and 0.5 mol/L NaCl onto a DEAE-52 cellulose column	[68]
*Cordyceps militaris*	Culture broth	EPS-III (A homogenous exopolysaccharide)	The culture broth was centrifuged, collected and concentrated; then it was precipitated with ethanol absolute (1:4); Sevage method and macroporous absorption resin (AB-8) to remove protein and pigment; further purified using Sephadex G-200 with distilled water	[69]
*Cordyceps militaris*	Fermentation broth	AEPS-II (An acidic exopolysaccharide)	The fermentation broth was centrifuged, concentrated and mixed with anhydrous ethanol; NKA-9 macroporous adsorption resin was used to remove pigment and protein using Sevage method; the crude EPS was isolated and purified with DEAE-Sephacel and Sephadex G-200 column chromatography, respectively	[45]
*Poria cocos*	Powder	PCP-1	The powder was extracted using deep eutectic solvent (ChoCl and oxalic acid in a molar ratio of 1:2); Sevage method was used for deproteinization; then the water solution was eluted on a Sephadex G-15 column	[71]
*Poria cocos*	Mycelial culture	FMGP	It was isolated from 49-day-old cultures of mycelia, after extraction with 0.1 M sodium acetate, centrifugation, precipitation and dialyzation, the supernatant was purified on a column of Fractogel BioSec	[72]
*Wolfiporia cocos*	Dried sclerotia	WIP (an acidic polysaccharide that is insoluble in water)	Dried sclerotia was extracted with NaOH solution (0.75 mol/L) and neutralized with HCl (1 mol/L); petroleum ether and hot water was applied to remove fat-soluble and water-soluble molecules; dialysis for removing inorganic salts	[73]
*Poria cocos*	Sclerotium	PCP-1C	The dried powder was extracted with ultrapure water and precipitated with ethanol; the Sevage method was used to remove protein; cellulose DEAE-52 column and Sephacryl S-500 column were applied to obtain PCP-1C	[74]
*Poria cocos*	Fermentation broth	EPS-0 M, EPS-0.1 M (exopolysaccharide)	Directly concentrated the supernatant of the fermentation broth; the water-soluble solution was dealt with using DEAE-52 cellulose anion exchange column and Sephadex G-100 gel column	[75]
*Poria cocos*	Lyophilized mycelium	IPS-0 M, IPS-0.1 M (intracellular polysaccharide)	Extract the lyophilized mycelium in hot water; the water-soluble solution was dealt with using DEAE-52 cellulose anion exchange column and Sephadex G-100 gel column	[75]
*Polyporus umbellatus*	Fruiting body	HPP	The Sevage method was used to remove protein; then the water solution was eluted using DEAE-52 cellulose column and Sephadex G-100 gel-filtration column, respectively	[76]
*Polyporus umbellatus*	Sclerotia	PUP-W-1	Boiling water extraction; initial separation was completed using DEAE-Sepharose Fast-Flow column with water and different concentrations of NaCl; further separation was completed via Superdex^TM^ G-75 column	[77]
*Polyporus grammocephalus*	Fruit body	PGPS	The fruit bodies were boiled with 4% NaOH and precipitated with ethanol; the crude water soluble polysaccharide was fractionated with GPC on Sepharose-6B column	[78]
*Omphalia lapidescens*	Fruit body	OL-2	The fruit body was extracted with hot water; the insoluble material was extracted with 0.1 M NaOH and 0.5 M NaOH, respectively; the 0.5 M NaOH soluble material was washed with water and 0.1 M NaOH, dissolved with 0.5 M NaOH and acidified with AcOH	[79]
*Lasiosphaera fenzlii*	Fruit body	TFP-1, TFP-2, TFP-3, TFP-4	The fruit body was extracted with 0.2 M NaOH solution; trichloroacetic acid was used to remove free protein; DEAE cellulose column, Sephacryl^TM^ S-200 and Sephacryl^TM^ S-300 gel columns were used for further isolation and purification	[80]
*Calvatia geigantea*	N/A	CGP I-1	Water extraction; DEAE-Sepharose fast flow ion-exchange column chromatography and Sephacryl S-300 gel filtration were used for isolation and purification	[81]

## 3. Structural Identification of Polysaccharides from Fungal TCM

According to literature reports, a wide range of natural polysaccharides and their derivatives from fungal TCM have been obtained using multiple extraction and purification strategies. Additionally, with limited methods and techniques, researchers have made a breakthrough in characterizing the chemical structures of those polysaccharides in multiple dimensions. Nonetheless, the current progress about the proposed structures of fungal TCM polysaccharides is summarized and listed in Table 3. 

### 3.1. Ganoderma

In a recent study by Zhao et al. [82], *Ganoderma lucidum* spore powder was fermented with Lactiplantibacillus plantarum ATCC14917. The structures of polysaccharides derived from *Ganoderma lucidum* before and after fermentation were analyzed. The results exhibited that the average molecular weight, monosaccharide composition, the content of uronic acid and the apparent structures has been significantly changed. Wen et al. [83] found that the water-soluble polysaccharides of *Ganoderma lucidum* were composed mainly by β-glucan and arabinogalactan. Chen et al. [60] obtained an acidic β-glucan (GLPC2) with hepatoprotective activity from *Ganoderma lucidum* fruiting body. Its average molecular weight was 20.56 kDa, and the further analysis showed that GLPC2 was mainly composed of D-Glcp-(1→, →3)-D-Glcp-(1→, →4)-D-Glcp-(1→, →6)-D-Glcp-(1→, →3,6)-D-Glcp-(1→, and→4)-D-GlcpA-(1→. A fucoxylomannan (FXM), with the molecular weight of 35.9 kDa, and was isolated and identified from *Ganoderma lucidum* by a team from Brazil [61]. The results showed that the main chain of FXM was α-D-Manp-(1→4)-linked units, and some of them were branched at O-6 position with α-L-Fucp-(1→2)-β-D-Xylp groups. Liu et al. [34] adopted cut-off ultrafiltration membrane combined with Sephacryl S-500 HR column to obtain a polysaccharide (RGLP-1) with a significant immunomodulatory activity. Furthermore, periodate oxidation, Smith degradation and methylation were combined to characterize the structure of RGLP-1. It was reported that RGLP-1 was composed of 1→3, 1→4, 1→6, and 1→3,6 glycosyl bonds in a ratio of 40.08:8.11:5.62:17.81. Sheng et al. [62] purified a polysaccharide (GLSP-I) with a molecular weight of 128 kDa from *Ganoderma lucidum* spore and characterized its structure with UPLC-MS/MS and NMR analysis. It was confirmed that the backbone of the polysaccharide was (1→3)-β-D-glucan, with side chains linking at O-6. Dong [63] et al. purified a selenized polysaccharide (SeMPN) from *Ganoderma lucidum* and identified its structure. HPGPC and monosaccharide composition results revealed that SeMPN was composed entirely of glucose with a molecular weight of 9.689 kDa. Further methylation analysis corroborated that SeMPN possessed three types of linkage, containing 1,4-linked Glcp, T-linked Glcp and 1,4,6-linked Glcp. Liu et al. [84] purified a bioactive β-D-glucan (GLSB50A-III-1) from water extracts of *Ganoderma lucidum* spores. The structural analysis results showed that the backbone of GLSB50A-III-1 was (1→3), (1→4), (1→6)-linked β-D-glucose, and the side chains consisted of β-(1→3) and β-(1→4)-linked residues, which were attached at O-6. 

### 3.2. Cordyceps

It is reported that a novel alkali-extracted polysaccharide, CM3-SII, is composed of →4)-β-D-Manp (1→, →6)-β-D-Manp (1→, and →6)-α-D-Manp (1→glycosyls, and has branching at the O-4 positions of →6)-β-D-Manp (1→glycosyls with β-D-Galp, (1→2) linked-β-D-Galf, and →2,6)-α-D-Manp (1→residues. And it was dominantly composed of mannose, glucose and galactose in a molar ratio of 10.6:1.0:3.7 [39]. Wanwan et al. [65] purified a homopolysaccharide (PACI-1), which composed of fructose, and the NMR spectrum showed that PACI-1 mainly contained β-configured pyranoside bonds. Unfortunately, the methylation analysis is deficient in this study, so the specific chemical structure of PACI-1 still remains unknown. Tang et al. [66] conducted differential scanning calorimetric analysis and helix–coil transition assay to confirm the structural feature of JCH-a1, which possessed a triple helix with more α-glycosides and had strong thermal stability. Additionally, JCH-a1 had a molecular weight of 60.7 kDa, with a total sugar content of 90.3% and a small amount of protein (2.32%). It was composed mainly of galactose, glucose and mannose with a molar ratio of 0.89:1:0.39. Wang et al. [67] identified the structural differences among the three polysaccharides from different parts of *Cordyceps cicadae* in terms of monosaccharide composition and methylation analysis. The specific parameters are shown in Table 3. Hu et al. [68] purified three new polysaccharides from *Cordyceps militaris*, and their total sugar contents were all higher than 90%. And their molecular weights were observed to be significantly higher than those of the polysaccharides that were reported previously. Their monosaccharide compositions were also more abundant than the previously reported polysaccharides. Interestingly, the three polysaccharides were found to be composed of glucosamine, which were isolated from *Cordyceps militaris for* the first time. Sun et al. [69] obtained a homogenous exopolysaccharide (EPS-III) from culture broth of *Cordyceps militaris* with Mw of 1.56 × 10^3^ kDa. Methylation analysis, Fourier infrared spectrum and NMR analysis were applied to characterized the structure of EPS-III. Meanwhile, it was found that EPS-III had helix structure when dissolved in weak alkaline solution and possessed branched and intertwined form on the surface. Additionally, a novel acidic exopolysaccharide (AEPS-II) was found in the fermentation broth of *Cordyceps militaris* [70]. It was confirmed that the molecular weight of the acidic pyranose was 61.52 kDa, which consisted of mannose, glucuronic acid, rhamnose, galactose acid, N-acetyl-galactosamine, glucose, galactose and arabinose.

### 3.3. Poria Cocos

Zhai et al. [71] obtained a glucan from *Poria cocos* (PCP-1), and the further structural characterization showed that PCP-1 might contain a triple helix structure. The putative structure of PCP-1 was determined as β-1, 3 glucan with a β-D-Glcp-(1→linkage connected to the main chain through an O(6) atom. Lin et al. [72] prepared a mannoglucan polysaccharide with gel filtration chromatography, which consisted of glucose, galactose, mannose and fucose in a molar ratio of 16:7:3:2. According to NMR analysis, the monomer polysaccharide was identified to be a highly branched 1, 3-β-mannoglucan. Yang et al. [73] identified a homogeneous polysaccharide with the molecular weight of 8.1 kDa from dried sclerotia of *Wolfiporia cocos*. It was a kind of pyranose form with β anomeric configuration, and the main chain was 1,3-β-glucan with amorphous structure. Cheng et al. [74] isolated a galactoglucan (PCP-1C) with a molecular weight of 17 kDa, which was composed of galactose, glucose, mannose and fucose. Further structural analysis results showed that the backbone of PCP-1C was 1,6-α-D-Glcp. Li et al. [75] characterized the four polysaccharides from different parts of *Poria cocos*, and they acquired the monosaccharide composition information with different molar ratios. Unfortunately, the sugar linkages and specific structure remains unknown due to the lack of methylation and NMR analysis in this study. 

### 3.4. Polyporus

Liu et al. [76] purified a new homogeneous glucan with the molecular weight of 6.88 kDa. The NMR results indicated that the homogenized polysaccharide has a backbone of 1,4-linked α-D-glucan with a (1→6)-α-D-glucopyranosyl side-branching unit. However, in the report of Gao et al. [77], the homogeneous polysaccharide was identified as (1,3), (1,6)-β-D-glucan with the molecular weight of 41.07 kDa. According to the acetylation status of PUP-W-1 hydrolytic monosaccharide residues, the polysaccharide was composed of only glucose. Sukesh et al. [78] confirmed that the glucan of *Polyporus* contained (1→3)-α-D-Glcp and (1→4)-α-D-Glcp moieties in a molar ratio of approximately 1:2. In conclusion, from the above three studies, we know that the researchers isolated different structures of glucans from *Polyporus*, which indicates that the glucan in *Polyporus* still has the potential to be developed, and it is necessary to further explore its novel chemical compounds and their biological activities.

### 3.5. Omphalia lapidescens

Rui [79] identified a glucan (OL-2) from *Omphalia lapidescens*, while the study only provided the results of NMR analysis. Specifically, it was confirmed that OL-2 consisted of a 1,3-β-glucan backbone chain decorated with a single six-branched β-glucosyl side unit on every fourth residue. However, the molecular weight of the glucan was still unclear. 

### 3.6. Lasiosphaera fenzlii

Through the literature review, Xia [80] has reported the isolation and identification of four homogeneous polysaccharides from *Lasiosphaera fenzlii* with molecular weights of 500 kDa, 600 kDa and 1000 kDa, respectively. These four polysaccharides had the same monosaccharide composition, but the molar ratio was different. Moreover, the results of methylation analysis and NMR analysis characterized the different structures of the four polysaccharides. Some researchers [81] have isolated a polysaccharide from another variety of *Lasiosphaera fenzlii*, *Calvatia geigantea*, with the monosaccharide composition of glucose, mannose and galactose.

## 4. Biological Functions 

### 4.1. Anti-Tumor Activity

The anti-tumor activities of polysaccharides from fungal TCM were investigated using in vivo and in vitro models, as shown in Figure 2 and Table 4. Furthermore, the bioinformatics study of network pharmacology and molecular docking strategy can also offer help in the investigation of anti-tumor research. Qin et al. [85] adopted network pharmacology, molecular docking and in vitro experimental validation to elucidate the detailed molecular targets and signaling mechanisms of Pachyman (*Poria cocos* polysaccharides) in treating hepatocellular carcinoma. Liu et al. [76] conducted a study to investigate the anti-bladder cancer activity and the potential mechanisms of homogeneous *Polyporus* polysaccharide (HPP). The BBN bladder cancer rat model and tumor-associated macrophages were utilized to assess the polarization of macrophages induced by HPP and its anti-bladder cancer mechanism. The results illustrated that HPP has good therapeutic effects on bladder tumors and drives TAM polarization to improve the tumor inflammatory microenvironment via NF-κB/NLRP3 signaling pathway, which suggested that HPP might be a potential therapeutic agent for bladder cancer. Jia et al. [86] also investigated the effect of anti-bladder cancer under the treatment of HPP. The results showed that HPP could enhance the expression of pro-inflammatory factors including IL-1β, TNF-α and iNOS, and surface molecules in macrophages and then could polarize macrophages to M1 type. Meanwhile, the activated macrophages could inhibit the proliferation of bladder cancer cells, regulate the apoptosis, and prevent migration and epithelial–mesenchymal transformation (EMT). Additionally, JAK2/NF-κB signaling pathway was suppressed in activated macrophages. Qi et al. [87] investigated the anti-cancer effect and its mechanisms on human colon cancer cell line (HCT116) of *Cordyceps sinensis* polysaccharide (CSP). The results indicated that CSP significantly downregulated the expression of PI3K and phosphorylation level of Akt and mTOR and upregulated the expression of AMPKa and phosphorylation level of ULK1. Xu et al. [88] conducted a study to find the anti-cancer effect of *Cordyceps cicadae* polysaccharides extracted by pre-soaking ultrasonic water extraction. It was confirmed that the polysaccharides could block the cell cycle in the S phase and promote the apoptosis, inhibiting the expression of mRNA and protein related to cell cycle and apoptosis signaling pathway. Moreover, a liver cancer model of H22 tumor-bearing mice was conducted to determine the anti-cancer effect and underlying mechanism of wild *Cordyceps* polysaccharides. Multiple experimental means of molecular biological uncovered that the anti-tumor mechanism of the large molecular weight polysaccharide from wild Cordyceps may be related to improving immune function and promoting the apoptosis of tumor cells [89]. WSG, a glucose-rich and water soluble polysaccharide isolated from *Ganoderma lucidum*, exhibited an anti-tumor effect on Murine Lewis lung carcinoma cell (LLC1)-induced lung cancer mice. Furthermore, it was found that co-treatment with WSG and cisplatin showed a synergistic inhibitory action on the proliferation of lung cancer cells, but WSG could ameliorate the cytotoxic effect of cisplatin at the same time [90]. It was coincidental that WSG could also suppress tongue cancer through restraining EGFR-mediated signaling pathways and promoting the apoptosis of tongue cancer cells [91]. Li et al. [92] discovered that the polysaccharides derived from *Ganoderma lucidum* (GLPS) could notably ameliorate hepatocellular carcinoma by affecting macrophage polarization and activating MAPK/NF-κB signaling pathway.

### 4.2. Anti-Oxidant Activity

The anti-oxidant activity of fungal TCM polysaccharides was investigated by detecting their scavenging ability of superoxide anion radical, DPPH radical and hydroxyl radical. In a study of Zheng [93], it was reported that PCPP had significant anti-oxidant activity in vitro, and the scavenging ability and reducing power of DPPH radicals and ABST radicals are dose-dependent. Yin et al. [94] found that *Polyporus umbellatus* polysaccharides (PPS) had the stronger scavenging ability for DPPH free radicals than for hydroxyl free radicals, and it was dose-dependent in a certain range. Liu et al. [58] extracted three exopolysaccharides from the broth of the liquid fermentation of *Polyporus umbellatus* and demonstrated their significant anti-oxidant activities. Specifically, these three polysaccharides showed an obvious scavenging action on DPPH and other free radicals in a dose-dependent manner. 

### 4.3. Immunomodulatory Activity

Liu et al. [95] investigated the carboxymethyl pachymaran (CMP), prepared from *Poria cocos* polysaccharide by carboxymethylation, and immunomodulatory activity evaluated in vitro. The following results demonstrated that the CMP exerted a significant immunomodulatory activity by regulating the secretion of iNOS, TNF-α and IL-6, which was related to its molecular weight and monosaccharide composition. Yu et al. [70] verified that a new acidic exopolysaccharide (AESP-II) could markedly promote the proliferation of spleen T and B lymphocytes in mice with immune injury. The results of the further analysis showed that the levels of cytokines and immunoglobulin secreted by T and B lymphocytes were increased after AESP-II treatment, respectively. Additionally, the activation of the MAPK signaling pathway may also be the mechanism of the immunomodulatory effect of AESP-II. Ying et al. [96] found that cultured *Cordyceps sinensis* polysaccharides (CSP) exerted a protective effect on cyclophosphamide (Cy)-induced intestinal mucosal immunosuppression in mice. Generally, the inflammatory cytokines secretion and transcription factors production were stimulated, and TLRs and NF-κB pathways were upregulated after CSP treatment. In a study of 2022, three polysaccharides (CCSP-1, CCSP-2 and CCSP-3) were purified from the spores of *Cordyceps cicadae*, while CCSP-2 exhibited significant immunosuppressive effect in mice. Meanwhile, the underlying mechanism has been elucidated, which may be connected with enhancing macrophage phagocytic activity, stimulating splenocyte proliferation, improving natural killer cytotoxicity, improving bone marrow suppression, regulating the secretion of cytokines and immunoglobulins and regulating oxidative stress [97]. 

### 4.4. Hypolipidemic Activity

It was reported that Wang et al. [98] demonstrated significant hypolipidemic activity on nonalcoholic fatty liver disease (NAFLD) phenotypes and the alteration of metabolism in mice on high-fat diet. The PCP markedly reduced serum and hepatic lipid levels and altered metabolic pathways including fatty acid metabolism, bile acid metabolism and tricarboxylic acid cycle. Yu et al. [99] investigated the lipid-lowering effect of CM3-SII, an alkali-extracted polysaccharide from *Cordyceps militaris*, in a heterozygous low-density lipoprotein receptor (LDLR)-deficient hamster model for hyperlipidemia. The results exhibited that CM3-SII improved hyperlipidemia through modulating the expression of multiple molecules related to lipid metabolism and the gut microbiota. CM1, a polysaccharide purified from *Cordyceps militaris*, observably reduced plasma total cholesterol and triglyceride levels in LDLR^(+/−)^ hamsters. Results suggested that CM1 improved hyperlipidemia through the downregulation of the plasma level of apolipoprotein B48, modulating the expression of certain significant genes and proteins in liver and suppressing preadipocyte differentiation in 3T3-L1 cells [100]. Yu et al. [101] confirmed that selenium (Se)-rich *Cordyceps militaris* polysaccharides (SeCMP) of 200 mg/kg exhibited anti-hyperlipidemia effect via mitigating obese-induced inflammation and modulating gut microbiota significantly. Wang et al. [102] found the lipid-lowering effects of *Ganoderma lucidum* polysaccharides. The experimental evidence suggested that the underlying mechanism may be related to regulating oxidative stress and inflammation response, improving bile acids synthesis and lipid regulatory factors, and accelerating cholesterol transport. 

### 4.5. Hypoglycemic Activity

Diabetes mellitus is a significant chronic metabolic disorder, which has become increasingly prevalent around the globe. Oral antihyperglycemic agents and insulin injection remain the mainstay for diabetes treatment. Unfortunately, such therapies are always accompanied by some significant adverse side effects. Therefore, to develop the novel, effective and affordable alternative therapeutic agents for diabetes is a great concern. Sun et al. [69] purified a novel homogeneous exopolysaccharide (EPS-III) from culture broth of *Cordyceps militaris*, and its hypoglycemic effect on STZ-induced diabetic mice was investigated. The EPS-III significantly inhibited α-glucosidase in vitro in a dose-dependent manner. The results in vivo indicated that EPS-III could improve diabetes via reducing plasma glucose concentration, improving glucose tolerance, protecting immune organs and repairing dyslipidemia. A purified fraction (AEPSa), obtained from *Cordyceps militaris* polysaccharides [103], was reported to ameliorate high-fat diet and streptozotocin (STZ)-induced T2DM mice. And the underlying mechanism may be related to reshaping gut microbiota against the TLR4/NF-κB pathway to protect the intestinal barrier. Wang et al. [67] compared the hypoglycemic effect of the polysaccharides isolated from different parts of *Cordyceps cicadae* and investigated the underlying mechanism. The bacterium substance polysaccharides (BSP), spore powder polysaccharides (SPP) and pure powder polysaccharides (PPP) significantly increased glucose absorption and alleviated insulin resistance in HepG2 cells. Interestingly, SPP was the most effective, and the mechanism underlying the hypoglycemic effect of SPP was the activation of the PI3K/Akt signaling pathway to ameliorate insulin resistance. Lee et al. [104] also investigated the mechanism of hypoglycemic effect of *Cordyceps militaris* polysaccharides. The results indicated that the polysaccharides significantly decreased the levels of blood sugar and serum lipids, Furthermore, they could improve intestinal dysbiosis. Shao et al. [105] obtained a thermally stable and non-toxic heteropolysaccharide F31 with significant hypoglycemic function from *Ganoderma lucidum*. The results showed that the levels of blood glucose of type 2 diabetic mice could be reduced by 42.25% through regulating gut microbiota when the oral administration dose of F31 was 180 mg/kg. While in a separate report, researchers revealed the different mechanisms by which F31 tackled diabetes [106]. They found that the hypoglycemic action of F31 may be related to kidney protection and adipocyte apoptosis. 

### 4.6. Hepatoprotective Activity

The mechanisms involved in the hepatoprotective activity of polysaccharides from fungal TCM still remain unclear. Jiang et al. [107] investigated the hepatoprotective activity of active Poria cocos polysaccharide (PCP-1C) using an vivo animal model. The results showed that PCP-1C observably ameliorated alcohol-induced liver injury by inhibiting the TLR4/NF-κB signaling pathway and improved hepatocyte apoptosis by restraining the cytochrome P450 2E1 (CYP2E1)/reactive oxygen species (ROS)/mitogen-activated protein kinases (MAPKs) signaling pathway. Additionally, PCP-1C could repair the intestinal barrier and decrease lipopolysaccharide (LPS) leakage. Chen et al. [108] investigated the hepatoprotective effect of *Ganoderma lucidum* polysaccharide (GLP, 25 kDa) through CCl_4_-induced mouse and TGF-β1-induced HSC-T6 cellular models of fibrosis. Multiple experimental means and detection index were used to assess the anti-fibrosis effect and underlying mechanisms of GLP. The results suggested that GLP could target inflammation, apoptosis, cell cycle and ECM–receptor interactions to treat liver fibrosis. 

### 4.7. Modulation on Gut Microbiota

The water-insoluble polysaccharides (WIP) from *Wolfporia cocos* were concentrated and used on C57BL/6 mice [109]. It has been discovered that WIF ameliorated the hepatic inflammatory injury and fat accumulation through modulating of gut microbiota. The abundance of Lachnospiraceae was increased and, the ethanol-induced fungal overgrowth was inhibited after oral administration of WIP, which activated the PPAR-γ signaling pathway. Wei et al. [110] found that PCP could regulate intestinal flora structure through increasing the relative abundance of *Prevotella*, *Bacteroides* and *Sutteralla*, and decreasing the ratio of *Firmicutes*/*Bacteroidetes* and the relative abundance of *Morganella* in nutritionally obese rats. Ying et al. [96] confirmed that CSP could regulate gut microbiota in immunosuppressive model mice via recovering SCFAs levels, improving microbial community diversity and modulating the overall structure of gut microbiota. Huang et al. [111] found that the polysaccharides from *Cordyceps militaris* (CMP) played a crucial role in the anti-obesity effect by modulating the gut microbiota. Specifically, CMP significantly improved the high-fat diet-induced gut microbiota dysbiosis, increased the abundance of *Alloprevotella*, *Parabacteroides*, *Butyricimonas* and *Alistipes*, and decreased the abundance of *Negativebacillus*. Li et al. [112] found that combined *Ganoderma lucidum* polysaccharide (GLP) and ciprofloxacin therapy exhibited the synergistic effect to diminish the side effects resulting from the *Salmonella* infection. The two drugs were used together to modulate the gut microbiota, especially to elevate the abundances of probiotic bacteria including *Lachnospiraceae* NK4A136, *Ruminococcaceae* UGG-014, *Lactobacillus* and *Parabacteroides*. The polysaccharides from sporoderm-broken spores of *Ganoderma lucidum* (BSGLP) were demonstrated to have gut microbiota regulation effect in high fat diet-induced mice. The results revealed that BSGLP could improve gut microbiota dysbiosis and intestinal barrier function and promote short-chain fatty acid production and GPR43 expression [113]. 

### 4.8. Anti-Inflammatory Activity

As recorded in the literature, the polysaccharides from *Poria cocos*, *Genoderma* and *Cordyceps* exerted significant anti-inflammatory effect on model animals. Furthermore, atherosclerosis is a chronic inflammatory cardiovascular disease whose pathogenesis involves the proliferation of vascular smooth muscle cells and lipid infiltration [114]. For example, Li Weifeng et al. found that *Poria cocos* polysaccharides could ameliorate high-fat diet-induced arteriosclerosis through the inhibition of inflammation and blood lipid levels in ApoE^−/−^ mice [115]. As another example, the anti-atherosclerosis effect of *Cordyceps militaris*-derived polysaccharide (CM-1) was confirmed. It was found that CM-1 markedly prevented the formation of atherosclerotic plaques by improving lipids excretion and reducing lipogenesis and lipolysis [116]. In 2020, Liu et al. [117] found that *Poria cocos* polysaccharides (PPs) could alleviate chronic nonbacterial prostatitis significantly. The results showed that inflammation and oxidative stress were prevented, hormone production was regulated, gut microbiota was modified and the DNA methylome was remodeled. Moreover, PPs [118] were found to recover the gut microbiota by targeting *Ruminococcaceae* NK4A214 group to treat chronic nonbacterial prostatitis. In a study by this team in 2022 [119], it was reported that the chronic nonbacterial prostatitis in rats could be alleviated by metabolites of gut microbiota fermenting *Poria cocos* polysaccharide. The results suggested that *Parabacteroides*, *Fusicatenibacter* and *Parasutterella* may be the crucial bacteria in alleviating chronic nonbacterial prostatitis and their metabolites, PPs 7-ketodeoxycholic acid and haloperidol glucuronide, may be the signal molecules of the “gut-prostate axis”. Apart from that, it was found that treatment with Carboxymethylated *Poria* Polysaccharides (CMP) [120] improved ulcerative colitis in mice caused by dextran sulfate sodium (DSS). Tan et al. [121] reported that *Poria cocos* polysaccharides (PCP) in the methionine and choline deficient-diet-fed nonalcoholic steatohepatitis (NASH) C57BL/6 mice alleviated histological liver damage, impaired hepatic function and increased oxidative stress. Meanwhile, PCP could reshape the composition of intestinal bacteria and downregulate the expression of pathways associated with immunity and inflammation (CCL3, CCR1, TLR4, Cd11b, NF-κB and TNF-α). Additionally, the study of Ye et al. [122] confirmed that PCP could prevent the progression of NASH and protect the intestinal barrier integrity under a high-fat diet via inhibiting the pyroptosis of small intestinal macrophages. Li et al. [123] confirmed the anti-atherosclerotic effect of *Ganoderma lucidum* polysaccharides in high-fat diet-induced rabbits, and the potential mechanism may be related to ameliorating endothelial dysfunction and inflammatory polarization of macrophages. It is reported that *Ganoderma lucidum* polysaccharides could also attenuate acute pneumonia induced by LPS in mice. And the underlying mechanisms were as follows: preventing the infiltration of inflammatory cells and cytokine secretion, blocking NRP1 activation and improving pneumonocyte apoptosis and autophagy [124]. Furthermore, GLP was proved to play an important role in improving AOM/DSS-induced colitis and tumorigenesis. Mechanistically, GLP could modulate microbiota disorder, reduce endotoxemia, improve gut barrier function and inhibit the inflammation in macrophage RAW 264.7, intestinal HT-29 and NCM460 cells [23]. 

### 4.9. Other Activities

In addition to the above pharmacological effects, different fungal TCMs possess their own unique biological activities, as described below. Combining animal model in vivo and cell models in vitro, Zheng [93] found that PCPP exerts a good radiation protection effect. Specifically, PCPP could effectively ameliorate the damage of spleen and liver, and improve the damage of hematopoietic system by regulating erythrocytes, platelets and hemoglobin. Meanwhile, it could significantly decrease the degree of oxidative damage caused by radiation in mice, suggesting that it could be considered as a potential resource for the development of natural radiation protection agents. Jiang et al. [125] illustrated that PPS exerted significant antifibrotic effects on mice and myofibroblasts. Bai [126] found that the polysaccharides of *Lasiosphaera fenzlii* possessed an inhibitory effect on both Staphylococcus aureus and Escherichia coli. At high concentration, Staphylococcus aureus had the strongest inhibitory capacity, while Escherichia coli had weaker inhibitory capacity, and its inhibitory ability weakened with the decrease in concentration. Two polysaccharides (CPA-1 and CPB-2) [127] purified from *Cordyceps cicadae* protected high-fructose/high-fat diet-induced obesity and metabolic disorders rats by reducing serum and hepatic lipid profiles, liver function enzymes and pro-inflammatory cytokines, alleviating hepatic oxidative stress and ameliorating histological alterations. An α-pyranose with a molecular weight of 15.94 kDa, separated and purified from *Cordyceps militaris*, was reported to have anti-allergic asthma effects. The potential mechanisms may be related to improving inflammatory cytokine levels, ameliorating the histopathological damages in the lung and intestinal tissues, regulating the oxidative and inflammatory pathways, reversing gut dysbiosis and improving microbiota function in allergic asthma model animals [128]. Huang et al. [129] investigated the promising ability to protect mice from obesity of Cordyceps militaris polysaccharides. The multi-angle analysis suggested that the underlying mechanism may be related to alleviating obesity-induced hyperlipidemia and insulin resistance, ameliorating systematic inflammation and regulating obesity-induced gut dysbiosis. What is novel is that the acidic polysaccharides purified from *Codyceps militaris* (CMPB) is beneficial for learning and memory impairment in mice. It can improve the fatigue state of high-intensity swimming mice and regulate the Nrf2-related signaling pathway to reverse the learning and memory impairment [130]. Zhang et al. [26] found that the polysaccharide (GLP-1) from *Ganoderma lucidum* could improve cognitive impairment in mice. Flow cytometry, ELISA analysis and metabolomics analysis indicated that GLP-1 could activate the Foxp3+ Treg cells to secret IL-10 and TGF-β1, and regulate the disorder energy metabolism. Li et al. [131] investigated the pharmacodynamic effect and underlying mechanism of anti-depression activity of *Ganoderma lucidum* polysaccharide (GLP) in a chronic social defeat stress depression animal model. The results indicated that GLP was an excellent agonist of Dectin-1 to show the significant anti-depression effect with various beneficial mechanism, such as regulating the neuroimmune system and AMPA receptor function. Tian et al. [132] found that the polysaccharides with different molecular weights derived from *Ganoderma lucidum* could be developed as new functional foods in consideration of their gastric injury-preventive activity. The results indicated that GLPs above 10 kDa exhibited the optimal effect via regulating anti-oxidation, inhibiting inflammatory responses and reducing the level of histamine in serum. Xu et al. [133] found that *Ganoderma lucidum* could protect sepsis-induced cardiac dysfunction in mice by affecting inflammation, apoptosis and proliferation. Wu et al. [134] conducted a cachexia model induced by the combination of cisplatin plus docetaxel in mice to investigate the curative effect of *Ganoderma lucidum* polysaccharide (Liz-H). The results showed that after administration of Liz-H, cachexia mice exhibited significant improvement due to a downregulation of muscle protein degradation-related genes and an increase in myogenic factors. Furthermore, Liz-H could restore the disordered gut microbiota to normal levels. Wu et al. [135] found the protective effect of *Ganoderma lucidum* polysaccharide (GLP) in D-galactose-induced aging salivary secretion disorders mice. The mechanism analysis results showed that GLP could maintain a healthy salivary gland (SG) and prevent SG deficiency by upregulating the rhythm and aquaporins. Zhong et al. [136] demonstrated that the polysaccharide GLP5 from *Ganoderma lucidum* could suppress the proliferation of leukemic cells by accelerating apoptosis. Chen et al. designed an experimental acute colitis in mice to determine the potential function of polysaccharides from natural *Cordyceps sinensis*. It was revealed that the polysaccharides significantly reduced the contents of inflammatory factors, alleviated colon tissue damage, enhanced the formation of IgA-secretory cells and sIgA contents, and modulated the gut microbiota. Taken together, the above reports imply that the polysaccharides from *Cordyceps sinensis* possess promising potentials as novel anti-colitis drug [137]. Li Yuan et al. purified an N-glycosidic polysaccharide-peptide complex CMPS-80 from the fruiting body of *Cordyceps militaris*. Experimental evidence revealed that CMPS-80 signally improved formation of atherosclerotic lesions and plasma lipid profiles in apolipoprotein E-deficient mice, which indicated that CMPS-80 could be considered as a candidate drug for prevention of hyperlipidemia and atherosclerosis [138].

**Table 4 molecules-28-06816-t004:** The bioactivities and mechanisms of polysaccharides from fungal TCM.

Bioactivity	Compound Name	Source	Subjects	Dose	Effects and Mechanism	Reference
Anti-tumor activity	Pachyman	*Poria cocos*	HepG2 human liver cancer cell and network pharmacology	0, 25 and 50 μM	Pachyman exerted an anti-cancer activity by elevating the intracellular level of ALB protein and downregulating the cellular content of VEGFA protein	[85]
	HPP	*Polyporus*	BBN-induced Fischer-334 rats and RAW 264.7, TPH-1 and T24 cells	1, 10 and 100 μg/mL	HPP could inhibit bladder cancer in BBN-induced rats by ameliorating histological damages in bladder; improve the tumor inflammatory microenvironment by regulating TAM polarization and NF-κB/NLRP3 signaling pathway	[76]
	HPP	*Polyporus*	Phorbol myristate acetate-induced THP-1 human leukemic cell	1, 10 and 100 μg/mL	HPP could confront bladder cancer through inhibiting the proliferation and progression of bladder cancer by the polarization of macrophages to M1 type, downregulating the JAK2/NF-κB signaling pathway	[86]
	CSP	*Cordyceps sinensis*	HCT116 cell line	0–800 μg/mL	CSP could inhibit the proliferation of HCT116 cells by inducing apoptosis and autophagy flux blockage. It might be achieved by modulating PI3K-Akt-mTOR and AMPK-mTOR-ULK1 signaling pathways	[87]
	CCP	*Cordyceps cicadae*	Hela cells	0, 25, 50, 100, 200, 400, 800 and 1600 μg/mL	CCP could inhibit the expression of Cyclin E, Cyclin A and CDK2, promote the expression of P53, activate Caspase cascade reaction, and up-regulate death receptor and the ratio of pro-apoptotic factor/anti-apoptotic factors to cause the cell cycle arrest and induce the apoptosis	[88]
	WCP	Wild *Cordyceps*	H22 tumor-bearing BALB/c mice	100 and 300 mg/kg	The large molecular weight polysaccharide could inhibit the proliferation of H22 tumors by improving immune function and promoting the apoptosis of tumor cells, and mainly interfering with IL-10/STAT3/Bcl2 and Cytoc/Caspase8/3 signaling pathways	[89]
	WSG	*Ganoderma lucidum*	LLC1 cells induced lung cancer C57BL/6 mice	75 mg/kg	WSG significantly prevented tumor growth and the formation of metastatic nodules in the lung tissue, and promoted the apoptotic responses mediated by cisplatin	[90]
	WSG	*Ganoderma lucidum*	Human tongue cancer SAS and HSC3 cells	0–800 μg/mL	WSG increased subG1 and G2/M populations and elevated Bax/Bcl2 ratio to induce apoptosis; inhibited phosphorylation of EGFR and AKT	[91]
	GLPS	*Ganoderma lucidum*	Mouse RAW 264.7 macrophages and hepatocellular carcinoma cell line Hepa1–6	0–200 μg/mL	GLPS markedly prevented the growth of Hepa1–6 allograft; promoted the expression of M1 phenotype marker CD86, iNOS, and pro-inflammatory cytokines (IL-12a, IL-23a, IL-27 and TNF-α); blocked macrophage polarization towards the M2 phenotype; reduced the expression of CD206, Arg-1, IL-6 and IL-10; upregulated the phosphorylation of MEK and ERK, IκBα and P65	[92]
Anti-oxidant activity	PCPP	*Poria cocos* peels	In vitro	1–5 mg/mL	PCPP has great anti-oxidant activity by scavenging DPPH radicals and reducing ABST radicals in a dose-dependent fashion	[93]
	PPS	*Polyporus umbellatus*	In vitro	1–8 mg/mL	PPS has the significant scavenging ability of DPPH free radicals and hydroxyl free radicals	[94]
	PPS	*Polyporus umbellatus*	In vitro	0.5–8 mg/mL	PPS exhibits the significant scavenging ability on DPPH and other free radicals in a dose-dependent manner	[58]
Immunomodulatory activity	CMP	*Poria cocos*	RAW 264.7	12.5, 25, 50, 100, 200 and 400 μg/mL	CMP plays a crucial role in immunoregulation by improving the secretions of iNOS, TNF-α and IL-6 through increasing the expression of iNOS, TNF-α and IL-6 mRNA	[95]
	AESP-II	*Cordyceps militaris*	Cyclophosphamide-induced BALB/c mice	25, 50 and 100 mg/kg	AESP-II could promote the proliferation of spleen T and B lymphocytes, increase the levels of cytokines and immunoglobulin secreted by T and B lymphocytes, and activate the MAPK signaling pathway to involve in the immunomodulatory function	[70]
	CSP	Cultured *Cordyceps sinensis*	Cyclophosphamide-induced female BALB/c mice	25, 50 and 100 mg/kg	CSP inhibited immunosuppression in mice via stimulating cytokines secretion (IL-12, IFN-γ, IL-4, IL-13, IL-6, IL-10, IL-17, TGF-β3, TNF-α, IL-2, IL-21) and transcription factors production (T-bet, GATA-3, RORγt, Foxp3), upregulating TLRs and NF-κB pathway key proteins	[96]
	CCSP-2	*Cordyceps cicadae*	Cyclophosphamide-induced immunosuppressive C57BL/6 mice	50, 100 and 200 mg/kg	CCSP-2 significantly increased spleen and thymus indices, enhanced macrophage phagocytic activity, stimulated splenocyte proliferation, improved natural killer cytotoxicity and bone marrow suppression, regulated the secretion of cytokines and immunoglobulins and modulated antioxidant enzyme system	[97]
Hypolipidemic activity	PCP	*Poria cocos*	High-fat diet-induced mice	1.5 g/day	PCP significantly reduced serum and hepatic lipid levels, and altered metabolic pathways including fatty acid metabolism, bile acid metabolism and tricarboxylic acid cycle	[98]
	CM3-SII	*Cordyceps militaris*	Heterozygous low-density lipoprotein receptor (LDLR)-deficient hamster	25, 100 mg/kg	CM3-SII attenuated total plasma cholesterol, non-high-density lipoprotein cholesterol and triglyceride; enhanced the concentration of plasma apolipoprotein A1 and the expression of liver X receptor α/ATP-binding cassette transporter G8 mRNA pathway and suppressed the expression of Niemann-Pick C1-like 1; downregulated sterol regulatory element-binding protein 1c and upregulated peroxisome proliferator-activated receptor α; increased the abundance of *Actinobacteria* and *Faecalibaculum* and the ratio of *Bacteroidetes/Firmicutes*.	[99]
	CM1	*Cordyceps militaris*	3T3-L1 cell; LDLR^(+/−)^ hamsters	100 μg/mL; 100 mg/kg	CM1 alleviated hyperlipidemia by downregulating the plasma level of apolipoprotein B48, modulating the expression of key genes and proteins in liver, small intestine and epididymal fat, and inhibiting preadipocyte differentiation in 3T3-L1 cells by suppressing the key genes involved in lipid droplet formation	[100]
	SeCMP	*Cordyceps militaris*	High-fat diet-fed C57BL/6 mice	50, 100 and 200 mg/kg	SeCMP-200 showed significantly hypolipidemic activity by decreasing serum triglyceride and low-density lipoprotein cholesterol, ameliorating obese-induced inflammation, decreasing the abundance of *Dorea*, *Lactobacillus*, *Clostridium*, *Ruminococcus* and increasing mucosal beneficial bacteria *Akkermansia*	[101]
	GLP	*Ganoderma lucidum*	High-fat diet-induced Kunming mice	100, 200 and 400 mg/kg	GLP inhibited the body weight gain and excessive lipid levels, ameliorated tissue injury; activated Nrf2-Keap1 and suppressed NF-κB signaling pathway; facilitated cholesterol reverse transport by LXRα-ABCA1/ABCG1 pathway; promoted the expression of CYP7A1 and CYP27A1; inhibited intestinal FXR-FGF15 expressions	[102]
Hypoglycemic activity	EPS-III	*Cordyceps militaris*	STZ-induced diabetic KM mice	60, 120 and 225 mg/kg	EPS-III exerted significantly hypoglycemic effect through alleviating weight loss, reducing plasma glucose concentration, improving glucose tolerance, protecting immune organs and repairing dyslipidemia	[69]
	AEPSa	*Cordyceps militaris*	High-fat diet and STZ-induced C57BL/6 mice	400 mg/kg	AEPSa ameliorating diabetes through increasing *Allobaculum*, *Alistipes*, *Lachnospiracae_NK4A136_group* and *norank_f_Muribaculaceae* and decreasing *Enterococcus* and *Ruminococcus_torques_group*, inhibiting TLR4/NF-κB pathway	[103]
	SPP	*Cordyceps cicadae*	HepG2 cells and T2DM KM mice	100, 200 and 400 mg/kg	SPP significantly increased glucose absorption and alleviated insulin resistance in HepG2 cells; SPP exerted hypoglycemic effect through activating PI3K/Akt signaling pathway to reduce hepatic insulin resistance	[67]
	CMP	*Cordyceps militaris*	High-fat/high-sucrose diet-induced C57BL/6 mice	N/A	CMP played a crucial role in the hypoglycemic effect by promoting the population of next generation probiotic *Akkermansia muciniphila* in the gut	[104]
	F31	*Ganoderma lucidum*	High fat diet and STZ-induced type 2 diabetic Kunming mice	60 and 180 mg/kg	F31 markedly decreased Firmicutes and enhanced the abundance of Bacteroidetes. Specifically, F31 may ameliorate glucose, insulin resistance and inflammation by inhibiting the release of endotoxins into the circulation from intestine, carbohydrate fermentation in gut and activation of intestine–brain axis	[105]
	F31	*Ganoderma lucidum*	C57BL/c and db/db mice	N/A	F31 ameliorated hyperglycemia through different approaches: decreased adenosine, galactitol and glycerophosphocholine and increased arginine, proline, arachidonic acid, creatine, aspartic acid, leucine, phenylalanine and ornithine to protect kidney function; increased Caspase-3, Caspase-6 and Bax and inhibited Bcl-2 to promote apoptosis in epididymal fat; reduced mitochondrial membrane potential to induce adipocyte apoptosis	[106]
Hepatoprotective activity	PCP-1C	*Poria cocos*	Alcohol-induced C57BL/6N mcie	25, 50 and 100 mg/kg	PCP-1C exerted a hepatoprotective action by decreasing inflammatory factor release, inhibiting oxidative stress and apoptosis, and ameliorating intestinal barrier injury	[107]
	GLP	*Ganoderma lucidum*	C57BL/6 mice and rat HSC-T6 hepatic stellate cell line	150 and 300 mg/kg; 0, 1.25, 2.5, 5, 7.5 and 10 mg/mL	GLP dramatically ameliorated hepatic fibrogenesis and inflammation by TLR4/NF-κB/MyD88 signaling pathway; blocked HSCs activation by reducing collagen I and a-SMA expressions; suppressed cell cycle; induced S phase arrest; inhibited the ECM-receptor interaction-associated molecule expressions (ITGA6 and ITGA8); restrained TGF-β/Smad signaling pathway in mice; decreased TGF-β1, Smad2 and Smad3 phosphorylation and promoted Smad7 expression in HSC-T6 cells	[108]
Modulation on gut microbiota	WIP	*Wolfporia cocos*	Alcohol-induced C57BL/6 mice	1 g/kg	WIP significantly enhanced the ratio of Firmicutes to Proteobacteria, increased the abundance of Lachnospiraceae and inhibited the ethanol-induced fungal overgrowth. It activated the PPAR-γ signaling pathway and facilitated a hypoxic state that suppressed the overgrowth of fungi and Proteobacteria in the gut	[109]
	PCP	*Poria cocos*	High-fat diet-induced nutritionally obese SD rats	50, 100 and 200 mg/kg	PCP could regulate intestinal flora structure by increasing the relative abundance of *Prevotella*, *Bacteroides* and *Sutteralla*, and decreasing the ratio of *Firmicutes*/*Bacteroidetes* and the relative abundance of *Morganella*	[110]
	CSP	Cultured *Cordyceps sinensis*	Cyclophosphamide-induced female BALB/c mice	25, 50 and 100 mg/kg	CSP regulated gut microbiota through recovering SCFAs levels, improving microbial community diversity, modulating the overall structure of gut microbiota, increasing the abundance of probiotics (*Lactobacillus*, *Bifidobacterium* and *Bacteroides*) and decreasing pathogenic bacteria (*Clostridium* and *Flexispira*)	[96]
	CMP	*Cordyceps militaris*	High-fat diet-induced C57BL/6 mice	200 and 400 mg/kg	CMP significantly improved the high-fat diet-induced gut microbiota dysbiosis, increased the abundance of *Alloprevotella*, *Parabacteroides*, *Butyricimonas* and *Alistipes*, and decreased the abundance of *Negativebacillus*	[111]
	GLP	*Ganoderma lucidum*	C57BL/c mice	N/A	GLP elevated the abundances of probiotic bacteria including *Lachnospiraceae* NK4A136, *Ruminococcaceae* UGG-014, *Lactobacillus* and *Parabacteroides*.	[112]
	BSGLP	Sporoderm-broken spores of *Ganoderma lucidum*	C57BL/6J mice	100 and 300 mg/kg	BSGLP improved gut microbiota dysbiosis; maintained intestinal barrier function; promoted short-chain fatty acid production and GPR43 expression; inhibited serum lipopolysaccharide level; augmented ileum expression of tight junction proteins and antimicrobial peptides; inhibited TLR4/MyD88/NF-κB signaling pathway in adipose tissue	[113]
Anti-inflammatory activity	PCP	*Poria cocos*	Arteriosclerosis in ApoE^−/−^ mice	100, 200, 400 mg/kg	The serum inflammatory mediators and lipids were inhibited; the pathological changes of the aorta were improved and the activation of TLR4/NF-κB pathway of the aorta was inhibited	[115]
	PPs	*Poria cocos*	Chronic nonbacterial prostatitis in SD rats	100, 250, 500 mg/kg	PPs plays the role of anti-chronic nonbacterial prostatitis via alleviating inflammation and oxidative stress, regulating hormone production, modifying gut microbiota and remodeling the DNA methylome	[117]
	PPs	*Poria cocos*	Chronic nonbacterial prostatitis in SD rats	250 mg/kg	PPs alleviates the chronic nonbacterial prostatitis by improving the histological damages in the inflamed prostate, inhibiting inflammation and regulating the gut microbiota by targeting *Ruminococcaceae* NK4A214 group	[118]
	PPs fermentation broth	*Poria cocos*	Chronic nonbacterial prostatitis in SD rats	250 mg/kg	It is proved that the metabolites of PPs 7-ketodeoxycholic acid and haloperidol glucuronide may be the signal molecules of the “gut-prostate axis”	[119]
	CMP	*Poria*	Ulcerative colitis in ICR female mice	300 mg/kg	CMP alleviated ulcerative colitis in mice through inhibiting colonic shortening and inflammation in colonic tissues, and regulating gut microbiota	[120]
	PCP	*Poria cocos*	Nonalcoholic steatohepatitis in C57BL/6 mice	150 and 300 mg/kg	The mechanism of PCP in preventing the development of NASH may be associated with the modulation of intestinal microbiota and the downregulation of the NF-κB/CCL3/CCR1 axis	[121]
	PCP	*Poria cocos*	Nonalcoholic steatohepatitis in C57BL/6 J mice and zebrafish	50, 100 and 200 mg/kg	PCP could slow down weight gain, hyperlipidemia and liver steatosis induced by high-fat diet; reduce the destruction of the gut-vascular barrier and the translocation of endotoxins; inhibit intestinal pyroptosis by regulating PARP-1	[122]
	CM1	*Cordyceps militaris*	Low-density lipoprotein receptor knockout (LDLR^−/−^) mice	25, 50 and 100 mg/kg	CM1 could reduce plasma lipid level and formation of atherosclerotic plaques through multiple pathways, enhanced plasma level of apolipoprotein A-I, decreased the levels of triglyceride, apolipoprotein B and total cholesterol, inhibited sterol regulatory element binding protein 1c, increased the liver X receptor α/ATP-binding cassette G5 pathway, inhibited PPAR-γ and adipose triglyceride lipase in epididymal fat	[116]
	GLPs	*Ganoderma lucidum*	High-fat diet-induced Japanese big-ear white rabbits	300 mg/kg	GLPs could prevent the progression of atherosclerosis through improving endothelial dysfunction and inflammatory polarization of macrophages, accelerating the apoptosis of foam cells	[123]
	GLP	*Ganoderma lucidum*	AOM/DSS-induced C57BL/6 mice	200 and 300 mg/kg	GLP ameliorated microbiota dysbiosis; promoted short-chain fatty acid production; inhibited TLR4/MyD88/NF-κB signaling pathway; increased numbers of goblet cells, MUC2 secretion, tight junction protein expressions; inhibited macrophage infiltration and IL-1β, iNOS, COX-2 expressions; inhibited the activation of MAPK	[23]
Other activities	PCPP	*Poria cocos peels*	AML-12 liver cell/^60^Co-γ induced KM mice	50–400 μg/mL/5, 10 and 20 mg/kg	PCPP exerts a significantly radiation protection effect through ameliorating the damage of spleen and liver, improving the damage of hematopoietic system by regulating erythrocytes, platelets and hemoglobin and decreasing the degree of oxidative damage	[93]
	PPS	*Polyporus*	Bleomycin-induced lung fibrosis C57BL/6 mice and human lung fibroblasts cell line	100 mg/kg; 1 mg/mL	PPS significantly improved bleomycin-induced lung fibrosis in mice through ameliorating pathological damages of lung tissues; it exerted antifibrotic effects in vitro via inhibiting fibroblast-to-myofibroblast transition, suppressing ECM deposition, repressing lung fibroblast proliferation and migration, suppressing TGF-1β-induced Smad2/3 activating	[125]
	RLSP	*Lasiophaere fenzlii*	In vitro	0.1, 0.01 and 0.001 mg/mL	RLSP possessed an inhibitory effect on both Staphylococcus aureus and Escherichia coil	[126]
	CPA-1 and CPB-2	*Cordyceps cicadae*	High fructose/high fat diet induced obesity and metabolic disorders rats	100 and 300 mg/kg	These two polysaccharides regulated metabolic disorders through inhibiting insulin and glucose tolerance, serum insulin and glucose levels, reducing serum and hepatic lipid profiles, liver function enzymes and pro-inflammatory cytokines, suppressing hepatic oxidative stress and hepatic lipid accumulation	[127]
	CMP	*Cordyceps militaris*	Ovalbumin-induced allergic asthma BALB/c mice	50, 100 and 200 mg/kg	CMP showed significantly anti-allergic asthma effects through improving inflammatory cytokine levels, ameliorating the histopathological damages, regulating oxidative and inflammatory pathways, reversing gut dysbiosis and improving microbiota function	[128]
	CMP	*Cordyceps militaris*	High-fat diet-induced C57BL/6 mice	400 mg/kg	CMP showed a promising ability to protect mice from obesity through ameliorating systematic inflammation, restoring the phylogenetic diversity of gut microbiota, increasing the relative abundance of short-chain fatty acid-producing bacteria, down-regulating the level of bacteria, which were positively related to the development of obesity	[129]
	CMPB	*Cordyceps militaris*	ICR mice	400 and 800 mg/kg	CMPB significantly decreased fatigue metabolites and oxidative stress, increased the expression level of BDNF, PI3K, Nrf2 and HO-1 in the hippocampus	[130]
	GLP-1	*Ganoderma lucidum*	Chronic cerebral hypoperfusion mice	N/A	GLP-1 improved cognitive impairment mice by elevating the levels of Foxp3+ Treg cell and inhibiting energy metabolism disorder	[26]
	GLP	*Ganoderma lucidum*	C57BL/6 mice and CD1 mice	1, 5 and 12.5 mg/kg	GLP inhibited the expression of IL-1β and TNF-α; promoted the expression of IL-10 and BDNF; prevented the activation of microglia and proliferation of astrocytes in hippocampus; increased the expression of GluA1 S845 phosphorylation as well as GluA1 and GluA2 expression levels	[131]
	GLPs	*Ganoderma lucidum*	Ethanol-induced acute gastric injury SD rat	100, 200 and 400 mg/kg	GLPs diminished the gastric injury in a dose-dependent manner through regulating anti-oxidation, inhibiting inflammation and decreasing the expression of histamine in serum	[132]
	GLP	*Ganoderma lucidum*	LPS-induced sepsis C57BL/6J mice	25 mg/kg	GLP could elevate the expression of SIRT1, decreased inflammatory factors in serum and inflammatory cells in heart tissues, blocked apoptosis and facilitated proliferation of myocardial tissues	[133]
	Liz-H	*Ganoderma lucidum*	Cisplatin plus docetaxel induced cachexia C57BL/6J mice	250 mg/mouse	Liz-H could block weight loss, muscle atrophy and neutropenia; downregulate muscle protein degradation-related genes (MuRF-1 and Atrogin-1); increase myogenic factors (MyoD and myogenin); restore the abundance of *Ruminococcaceae* and *Bacteroides* to normal levels	[134]
	GLP	*Ganoderma lucidum*	D-galactose-induced C57BL/6J mice	N/A	GLP increased the expression levels of AQP5, AQP4, AQP1; blocked the release of inflammatory factors; upregulated core clock genes and proteins; restored the co-localized expression of CLOCK and AQP5	[135]
	GLP	*Ganoderma lucidum*	LPS-induced C57BL/6N mice	25, 50 and 100 mg/kg	GLP inhibited inflammatory cell infiltration; reduced the expression levels of GM-CSF, IL-6, IL-1β, TNF-α and Saa3; blocked the activation of NRP1; promoted the expression of Bcl2/Bax and LC3; decreased the ratio C-Caspase 3/Caspase 3 and P62 expression	[124]
	GLP5	*Ganoderma lucidum*	Human acute T cell leukemia cell line	25 and 50 mg/L	GLP5 notably suppressed the proliferation of Jurkat cells; increased the expression levels of Caspase3; regulated the expression levels of Bax and Bcl-2	[136]
	CSP	*Cordyceps sinensis*	Dextran sodium sulfate-induced C57BL/6J mice	N/A	CSP significantly increased the colon length; improved colon tissue damage; inhibited the activation of NF-κB pathway; decreased the expressions of inflammatory cytokines; augmented the number of goblet cells; regulated the expressions of intestinal tight junction proteins (Occludin and Claudin-1); promoted the formation of IgA-secretory cells and sIgA contents	[137]
	CMPS-80	*Cordyceps militaris*	Apolipoprotein E-deficient mice	N/A	CMPS-80 dramatically blocked formation of atherosclerotic lesions and plasma lipid profiles; regulated multiple lncRNA-microRNA-mRNA axes	[138]

## 5. Quality Control

With the development of analytical techniques and the depth of polysaccharide research, innovative methods and rich detection means have greatly driven the all-side structural representation of Chinese medicine polysaccharides at multi-levels [139,140]. Under the background of clear structure and significant biological action of Chinese medicine polysaccharide, we may consider the macromolecular polysaccharide components as a standard to evaluate the quality of traditional Chinese medicine.

A novel two-dimensional correlation infrared spectrometry (2D COS-IR) method is established to rapidly discriminate the polysaccharides in traditional Chinese medicine. As recorded in the research, the constructed 2D COS-IR method combined with chemometric analysis has achieved the quick and effective identification and discrimination of polysaccharides in Chinese medicine. It provides an important methodological reference for us to evaluate the quality of traditional Chinese medicine polysaccharide [141]. In terms of the analysis strategy, the following studies have made outstanding contributions to the quality control of traditional Chinese medicine polysaccharides. Chen et al. [142] conducted the phytochemical properties of *Maidong* polysaccharide including yield, the content of fructose, molecular parameters and compositional monosaccharides as the indicators for quality control. The team investigated and compared 29 batches of *Maidong* polysaccharides from different species and origin using the mentioned analysis method, which was helpful to improve their quality control. Fingerprint profiling has been popularly applied for quality evaluation of plant polysaccharides in recent years. For instance, in a report, polysaccharides extracted from ten *Fritillaria* species were compared and distinguished through multi-levels evaluation strategy by researchers [143]. Fingerprint profiling technique was conducted to evaluate the quality of Fritillaria polysaccharides from the following four levels: (1) establish models based on infrared signals for identification of affinis and multi-origin species; (2) profile the oligosaccharide mapping after partially hydrolyzing and characterizing the structural information; (3) interpret the monosaccharide composition and quantify the main monosaccharide; and (4) establish a data fusion model according to the stretching vibration of functional groups, polymerization degree, oligosaccharides abundance and monosaccharide content. Similarly, in another report [144], holistic fingerprinting at polysaccharide and the hydrolyzed oligosaccharide and monosaccharide levels based on various chromatography methods was established to evaluate the quality of polysaccharides from six root ginseng drugs. Accordingly, it can be seen that fingerprint technique combined with chemometry has become a popular and reliable method to evaluate the quality of traditional Chinese medicine polysaccharide. Sun et al. [145] listed polysaccharide molecular weight, monosaccharide composition and total polysaccharide content as evaluation indexes, combined with small molecules qualitative and quantitative analysis results, and holistically evaluated the quality of *Callicarpae Formosanae Folium*, which provided a novel idea for scholars to evaluate the quality of traditional Chinese medicine. The same analytical strategy has been applied to evaluate the quality of *jujube* polysaccharide [146]. Additionally, gas–liquid chromatography and mass spectrometry are also an effective means to evaluate the quality of polysaccharide. Xia et al. [147] developed structural fingerprinting of polysaccharides with monosaccharide compositional fingerprinting, Smith degradation and non-degradation fingerprinting, and oligosaccharide compositional fingerprinting based on gas–liquid chromatography and mass spectrometry to discern *Panax* species. The approach possesses high comprehensibility and satisfactory generalization capability for analysis and quality evaluation of plant polysaccharides.

*Ganoderma lucidum* is loved and praised by the larger masses because of its abundant nutritional value and medicinal properties. Taking into consideration the existence of fake and confused *Ganoderma lucidum* products on the market, it is essential to establish reasonable quality standards. As recorded in China Pharmacopoeia of 2020 edition, polysaccharide is considered as a content determination index to evaluate the quality of *Ganoderma lucidum* [148]. Furthermore, an acidic hydrolysate fingerprints based on HILIC-ELSD/MS combined with multivariate analysis strategy was built to investigate the quality of *Ganoderma lucidum* polysaccharides [149]. On the one hand, the method was applied to evaluate the *Ganoderma lucidum* samples from different cultivation bases, and polysaccharides and D-galactose were considered as candidate biomarkers. On the other hand, the intraspecific differentiation was preliminarily surveyed, in which L-rhamnose, D-xylose, L-arabinose and mannose were selected as potential chemical markers. The research suggested great potential in the quality control of *Ganoderma lucidum*. Wang et al. [150] determined the weight average relative molecular weight of *Ganoderma lucidum* polysaccharides with gel permeation chromatography, differential refraction detector and multi-angle laser scattering. The monosaccharide composition of different batches of *Ganoderma lucidum* polysaccharides was analyzed, which provided technical reference for establishing the quality control of polysaccharides in *Ganoderma lucidum* extract. Liu [151] et al. employed chemometric method combined with multiple fingerprints to assess the immunomodulatory polysaccharides from *Ganoderma lucidum*. The results indicated that the polysaccharides with molecular weight of 4.27 × 10^3^~5.27 × 10^3^ and 1 × 10^4^~1.14 × 10^4^ g/mol were the main active fractions. Furthermore, galactose, mannose, glucuronic acid and some oligosaccharide fragment also dramatically affected the immune activity of polysaccharides. And these ingredients associated with immunomodulatory activity could be considered as important markers to assess the quality of *Ganoderma lucidum* polysaccharides. Zha [37] conducted research to evaluate the quality of *Ophioordyceps sinensis* in different areas of Qinghai Province based on polysaccharide. In this study, the best polysaccharide extraction technology was applied to extract and compare the yield of polysaccharide from the mycelia of *Cordyceps sinensis* in different regions of Qinghai Province. And the yield of polysaccharide was considered as an indicator to evaluate the quality. Zhou et al. [152] conducted a study on comparing the difference in polysaccharide content, molecular weight distribution and monosaccharide composition of big grass and little grass of *Cordyceps sinensis*. The investigation showed the significant difference between the samples, which provided data support for traditional commodity classification and quality control of *Cordyceps sinensis*. Zhang et al. [153] established multiple fingerprints to evaluate the quality of polysaccharides from *Poria cocos*. HPGFC-ELSD (high performance gel filtration chromatography-evaporative light scattering detector) fingerprint, PMP-HPLC-DAD (1-phenyl-3-methyl-5-pyrazolone-high performance liquid chromatography-diode array detector) fingerprint of complete acid hydrolysates and HILIC-HPLC-ELSD (hydrophilic interaction-high performance liquid chromatography-evaporative light scattering detector) fingerprint of enzyme hydrolysates combined with chemometrics methods were conducted in this study. The results showed that although 16 batches of *Poria cocos* polysaccharides from different regions showed high similarity in structural characteristics, two commercial samples were still authenticated to be adulterants. It provided a powerful experimental basis for polysaccharides to be considered as quality evaluation index of *Poria cocos*. Yi et al. [154] used near infrared spectrometry combined with chemometrics to rapidly determine the polysaccharide contents of *Poria cocos* under the optimal procedures. The developed effective and feasible method showed good applicable value for rapid quality evaluation of *Poria cocos*. Similarly, Xie et al. [155] also adopted the above method to evaluate the quality of *Poria cocos* from multiple cultivation regions, which further completed the quality evaluation system. Zhu et al. [156] performed a comparative study on qualitative and quantitative characterization of polysaccharide profiles in three different medicinal parts of *Poria cocos*. They found that arabinose, glucose, galacturonic acid and ribose played crucial roles in the clusters between the three parts. Such differences in the chemical components in the three parts could not only explain the clinical application, but also provide a chemical basis for the quality control of *Poria cocos*. Song et al. compared the polysaccharide content, monosaccharide composition and monosaccharide content of *Grifola* from different regions [157]. They proved the existence of different polysaccharide contents between multiple regions, which provided experimental evidence for quality evaluation of *Grifola*. Guo et al. [158] built a qualitative and quantitative method to analyze the monosaccharide composition of *Polyporus umbellatus* polysaccharides based on high-performance liquid chromatography coupled with electrospray ionization-ion trap-time of flight-mass spectrometry (HPLC-ESI-TOF-MS). The accurate method with good recovery can be applied to the quality control of *Polyporus umbellatus*.

## 6. Discussion and Perspectives

In the history and experience of the application of traditional Chinese medicine, fungal traditional Chinese medicine plays a crucial role in maintaining human health and treating diseases. As listed in the Chinese Pharmacopoeia of 2020 edition, the quality standards of six kinds of fungal TCM are mainly included. Additionally, polysaccharide is one of the dominant pharmacodynamic materials and has attracted increasing attention because of its multiple biological functions according to literature reports. With this background, the current review established the research progress on fungal TCM polysaccharides in the past three years, and we comprehensively summarized the newly available information in terms of extraction, purification, structural identification, biological functions and quality control. Nevertheless, there are still some challenges and problems that have to be solved that need further research. First of all, in spite of the fact that multiple extraction and purification methods have been developed to obtain the polysaccharides from fungal TCM, a systematic and suitable process for industrial production is still lacking. Meanwhile, how to gather high-purity and highly active polysaccharides under the premise of cost control is also an important problem to be solved. Secondly, more precise chemical structure of fungal TCM polysaccharides need to be characterized. The chemical structures of polysaccharides of some fungal TCM with strong biological activity are still unknown. Thirdly, a large number of studies have been conducted to demonstrate the biological functional diversity of fungal TCM polysaccharides, but the mechanisms of their absorption, degradation, distribution and metabolization in vivo are unclear. Additionally, it is necessary to carry out toxicity tests and clinical studies for developing health foods from some fungal TCM polysaccharides with health benefits. Last but not the least, it is of great importance to establish a strict system to investigate the quality control of fungal TCM polysaccharides. To our best knowledge, different medicinal parts and different preparation processes will affect the content and structure of polysaccharide, which may further change its biological activity.

## 7. Conclusions

In conclusion, the polysaccharides from fungal TCM can exert broad spectrum biological functions in the fields of functional food additives and pharmaceuticals. Nevertheless, opportunities and challenges co-exist. Although the natural polysaccharides have the characteristics of being green, harmless, biodegradable and easy to prepare, the pharmacodynamic mechanism and quality control are still the focus of future study. It is effectively helpful for TCM to enter the international market and serve as an ideal choice for health care. The current review may provide readers with the latest research progress of fungal TCM polysaccharides, and further accelerate the understanding of the biological activity of fungal TCM polysaccharides and corresponding applications in multiple fields.

## Figures and Tables

**Figure 1 molecules-28-06816-f001:**
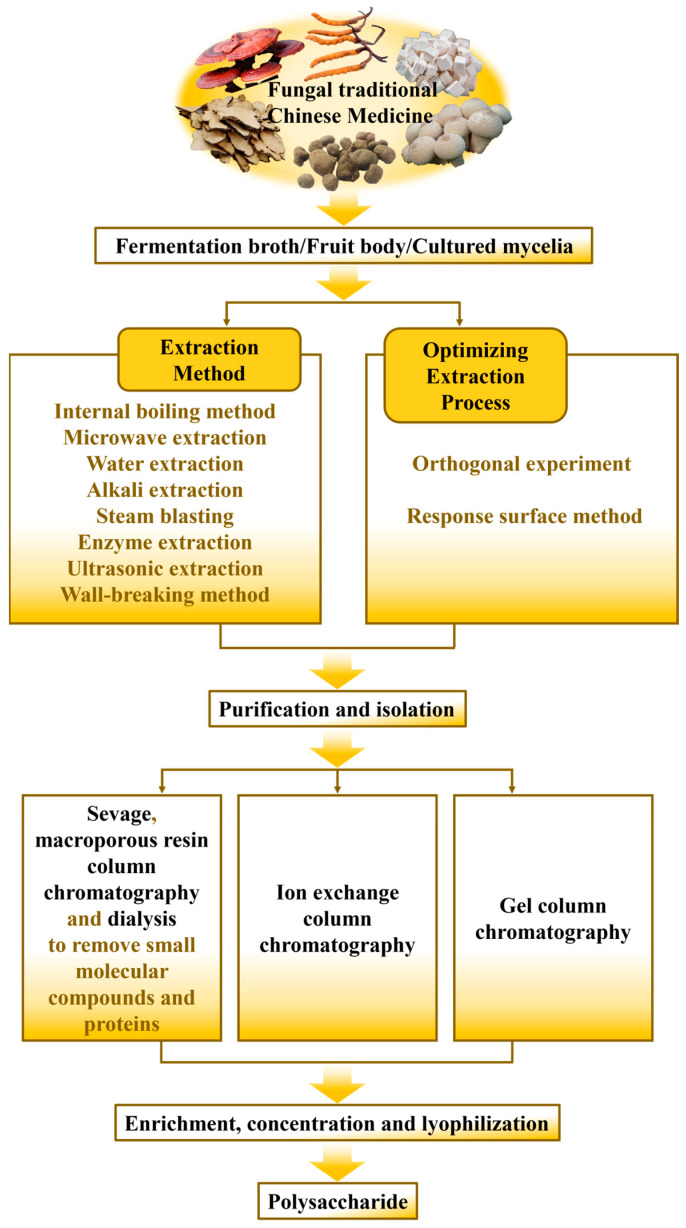
Extraction, isolation and purification processes of polysaccharides from fungal TCM.

**Figure 2 molecules-28-06816-f002:**
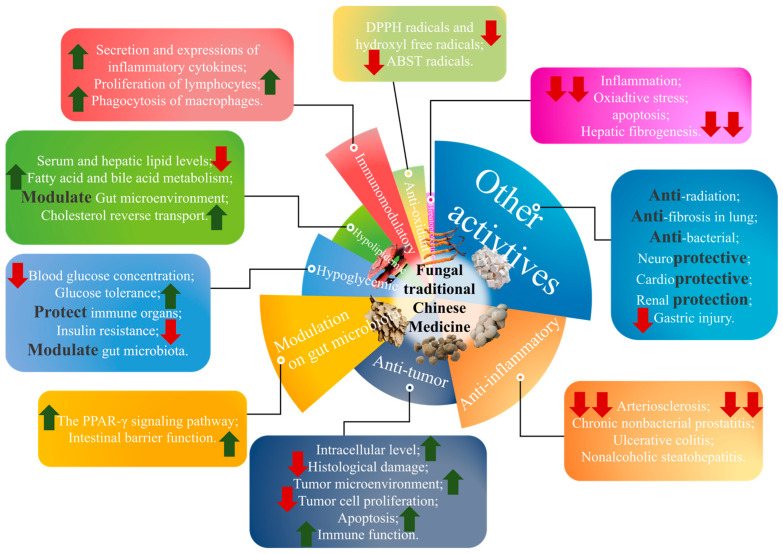
The biological functions of polysaccharides from fungal TCM.

**Table 3 molecules-28-06816-t003:** The structural characteristics of polysaccharides from fungal TCM.

Compound Name	Source	Molecular Weight	Monosaccharide Composition	Proposed Structure	Reference
GLP	*Ganoderma lucidum*	112 kDa	Xylose:mannose:galactose: glucose:glucuronic acid = 1.53:15.64:1.84:80.6:0.39	β-1,6-Glcp and β-1,3-Glcp linkages were the main ones that existed; the apparent structure was porous and loose	[82]
GLPC2	*Ganoderma lucidum* fruiting body	20.6 kDa	Mannose:glucuronic acid:glucose:galactose:xylose:fucose = 5.9:9.0:80.4:1.8:1.8:0.9	GLPC2 was mainly composed of D-Glcp-(1→, →3)-D-Glcp-(1→, →4)-D-Glcp-(1→, →6)-D-Glcp-(1→, →3,6)-D-Glcp-(1→, and→4)-D-GlcpA-(1→	[60]
FGLP	*Fermented Ganoderma lucidum*	88.9 kDa	Arabinose:xylose:mannose:galactose:glucose:glucuronic acid = 0.42:0.82:34.31:3.32:59.47:1.66	β-1,6-Glcp and β-1,3-Glcp linkages were the main ones that existed; the glucose content was decreased, and the uronic acid content was increased; the apparent structure was smooth and hard	[82]
GLSP	*Ganoderma lucidum*	4–5485 kDa	Arabinose:glucose:galactose = 1.23:90.82:7.95	The types of glycosidic linkages that existed in GLSP were as follows: β-Glcp-(1→; →3)-β-D-Glcp-(1→; β-Glcp-(1→or→6)-β-D-Glcp-(1→; →5)-α-Araf-(1→; →6)-α-Galp-(1→; →4)-α-Galp-(1→; →6)-α-Galp-(1→4)-α-Galp-(1→	[83]
FXM	*Ganoderma lucidum*	35.9 kDa	Fucose:xylose:mannose:glucose = 24:25:48.3:2.7	The main chain of FXM was α-D-Manp-(1→4)-linked units, and some of them were branched at O-6 position with α-L-Fucp-(1→2)-β-D-Xylp groups	[61]
RGLP-1	*Ganoderma lucidum*	3978 kDa	Fucose:mannose:glucose:galactose = 0.13:0.05:0.72:0.10	RGLP-1 contained 12 linkage forms: (1→3)-linked glucose; (1→)-linked glucose; (1→3,6)-linked glucose; (1→4)-linked glucose; (1→6)-linked glucose; (1→3)-linked galactose; (1→2,3)-linked galactose; (1→)-linked galactose; (1→6)-linked galactose; (1→4,6)-linked galactose; (1→3)-linked mannose; and (1→2)-linked mannose	[34]
GLSP-I	*Ganoderma lucidum* spore	128 kDa	Glucose	The backbone was (1→3)-β-D-glucan, and three side chains including Glc-(1→3)-Glc-(1→3)-Glc-(1→6)-Glc, Glc-(1→6)-Glc-(1→6)-Glc-(1→6)-Glc and Glc-(1→3)-Glc-(1→3)-Glc-(1→3)-Glc-(1→3)-Glc were linked at O-6	[62]
SeMPN	*Ganoderma lucidum* mycelia	9.7 kDa	Glucose	The backbone was 1,4-linked Glcp, and the other types of linkage contained T-linked Glcp and 1,4,6-linked Glcp	[63]
GLSB50A-III-1	*Ganoderma lucidum* spore	193 kDa	Glucose	The backbone of GLSB50A-III-1 was (1→3), (1→4), (1→6)-linked β-D-glucose, and the side chains consisted of β-(1→3) and β-(1→4)-linked residues, which were attached at O-6	[84]
CM3-SII	*Cordyceps militaris* fruiting body	25.2 kDa	Mannose:glucose:galactose = 10.6:1.0:3.7	→4)-β-D-Manp(1→, →6)-β-D-Manp(1→, and→6)-α-D-Manp(1→glycosyls, and branching at the O-4 positions of →6)-β-D-Manp(1→ glycosyls with β-D-Galp, (1→2) linked-β-D-Galf, and →2,6)-α-D-Manp(1→ residues, O-6 and O-2 positions of the →2,6)-α-D-Manp(1→ residues were substituted with methyl and β-D-Galp	[39]
PACI-1	*Cordyceps cicadae* fermentation medium	10 kDa	A homopolysaccharide composed of fructose	NMR spectrum indicated that PACI-1 mainly contained β-configured pyranoside bonds	[65]
JCH-a1	*Cordyceps cicadae* fruiting body	60.7 kDa	Galactose:glucose:mannose = 0.89:1.0:0.39	It has a triple helix with more α-glycosides and has strong thermal stability	[66]
BSP	*Cordyceps cicadae* bacterium substance	N/A	Arabinose:galactose:glucose:xylose = 7.60:1.80:76.10:14.50	Araf-(1→, Xylp-(1→, →5)-Araf-(1→, Manp-(1→, Galp-(1→, →3, 5)-Araf-(1→, →2)-Manp-(1→, →6)-Manp-(1→, →4, 6)-Glcp-(1→, →4, 6)-Galp-(1→, →2, 6)-Manp-(1→	[67]
SPP	*Cordyceps cicadae* spore powder	N/A	Arabinose:galactose:glucose:xylose:mannose = 9.10:15.40:41.20:17.50:16.80	Araf-(1→, →5)-Araf-(1→, Glcp-(1→, →3, 5)-Araf-(1→, →3)-Glcp-(1→, →4)-Glcp-(1→, →3, 4)-Glcp-(1→, →4, 6)-Glcp-(1→	[67]
PPP	*Cordyceps cicadae* fruiting body	N/A	Arabinose:galactose:glucose:xylose:mannose = 4.50:11.90:62.20:10.70:10.70	Araf-(1→, Xylp-(1→, →5)-Araf-(1→, Manp-(1→, Galp-(1→, →3, 5)-Araf-(1→, →2)-Manp-(1→, →3)-Glcp-(1→, →4)-Galp-(1→, →4)-Glcp-(1→, →6)-Glcp-(1→, →6)-Manp-(1→, →3, 4)-Glcp-(1→, →4, 6)-Glcp-(1→, →2, 6)-Manp-(1→	[67]
CMP-1	*Cordyceps militaris*	2.2 × 10^3^ kDa	Mannose:Glucosamine:Ribose:Rhamose:Glucuronic acid:Galacturonic acid:Glucose:Galactose:Xylose:Arabinose:Fucose = 39.35:4.03:3.98:2.56:1.62:1.52:70.52:26.90:1.00:3.23:4.23	It has three weak absorption peaks (1109 cm^−1^, 1125 cm^−1^ and 1151 cm^−1^) in the range of 1110~1160 cm^−1^, which may be contributed to side chains on α-carb\zon from the second derivative of infrared spectrum. And it has three broad absorption bands in the range of 990~860 cm^−1^	[68]
CMP-2	*Cordyceps militaris*	2.8 × 10^3^ kDa	Mannose:Glucosamine:Ribose:Rhamose:Glucuronic acid:Galacturonic acid:Glucose:Galactose:Xylose:Fucose = 13.62:86.70:7.80:6.22:1.47:2.99:17.80:9.26:1.00:3.02	It has two weak absorption peaks (1109 cm^−1^ and 1151 cm^−1^) in the range of 1110~1160 cm^−1^ and four absorption peaks in the range of 990~860 cm^−1^	[68]
CMP-3	*Cordyceps militaris*	1.7 × 10^3^ kDa	Mannose:Glucosamine:Ribose:Rhamose:Glucuronic acid:Galacturonic acid:Glucose:Galactose:Xylose:Fucose = 33.61:44.67:27.34:31.84:7.32:8.39:102.23:38.27:1.00:5.79	It has two weak absorption peaks (1109 cm^−1^ and 1125 cm^−1^) in the range of 1110~1160 cm^−1^ and six absorption peaks in the range of 990~860 cm^−1^	[68]
EPS-III	*Cordyceps militaris* culture broth	1.6 × 10^3^ kDa	Mannose:Glucose:Galactose = 1.68:1:1.83	The backbone was mainly consisted of →4)-α-D-Galp-(1→, while →3, 6)-α-D-Manp-(1→, →4)-α-D-Manp-(1→, →3)-β-D-Galp-(1→ and →3)-α-D-Glcp-(1→	[69]
AEPS-II	*Cordyceps militaris* fermentation broth	61.5 kDa	Mannose:Glucuronic acid:rhamnose:galactose acid:N-acetyl-galactosamine:Glucose:galactose:Arabinose = 1.07:5.38:1:3.14:2.23:15:6.09:4.04	The backbone contained →4)-α-D-Glcp-(1→, 4, 6-α-D-Glcp-(1→, 2, 4)-β-D-Glcp-(1→, →3)-α-D-Glcp-(1→, →3)-β-D-Glcp-(1→, →4)-β-D-Galp-(1→, →4)-α-D-GalpA-(1→, →4)-β-D-Glcp-(1→, →6)-β-D-GalNAc-(1→; the linkages of branches were mainly composed of 4, 6-α-D-Glcp-(1→, →3)-α-D-Araf-(1→, →2, 4)-β-D-GlcpA-(1→	[70]
PCP-1	*Poria cocos*	3.2 kDa	Nearly 100% glucose	The main linkages of backbone structure were 1, 3-linked glucose	[71]
FMGP	*Poria cocos* cultures mycelia	31.7 kDa	Glucose:Galactose:Mannose:Fucose = 16:7:3:2	The main skeleton was a 1,4-α-Man-interlaced-1,3-β-Glucan with interlaced 6-O-α-L-fucosyl 1,4-α-Glc and 1,4-α-Gal branches	[72]
WIP	*Wolfiporia cocos* dried sclerotia	8.1 kDa	Glucose	It was a kind of pyranose form with β anomeric configuration; the main chain was 1,3-β-Glucan with amorphous structure	[73]
PCP-1C	*Poria cocos* sclerotium	17 kDa	Galactose:Glucose:Mannose:Fucose = 43.5:24.4:17.4:14.6	The backbone consisted of 1,6-α-D-Glcp, whose branches composed of 1,3-β-D-Glcp, 1,4-β-D-Glcp, 1,6-β-D-Glcp, T-β-D-Glcp, T-α-D-Manp, T-α-L-Fucp, 1,3-α-L-Fucp	[74]
EPS-0M	*Poria cocos* fermentation broth	1.8 × 10^3^ kDa	Glucose:Mannose:Galactose:Fucose:Rhamnose = 17.3:46.3:19.9:8.7:5.0	It possessed β-type glycosidic bonds and exhibited loose fragmentary aggregation with an amorphous spherical structure	[75]
EPS-0.1M	*Poria cocos* fermentation broth	2.0 × 10^3^ kDa	Glucose:Mannose:Galactose:Fucose:Rhamnose = 11.5:46.5:21.9:10.7:5.6	It possessed β-type glycosidic bonds and exhibited loose fragmentary aggregation with an amorphous spherical structure	[75]
IPS-0M	*Poria cocos* mycelium	30 kDa	Glucose:Mannose:Galactose:Fucose:Rhamnose = 79.7:8.9:5.5:1.7:3.1	It possessed β-type glycosidic bonds and it was mainly filamentous and rod-like, with a relatively regular structure	[75]
IPS-0.1M	*Poria cocos* mycelium	5.0 × 10^3^ kDa	Glucose:Mannose:Galactose:Fucose:Rhamnose = 50.3:20.9:16.1:6.0:4.0	It possessed β-type glycosidic bonds and it was mainly filamentous and rod-like, with a relatively regular structure	[75]
HPP	*Polyporus umbellatus*	6.9 kDa	Only glucose	It has a backbone of 1,4-linked α-D-glucan with a (1→6)-α-D-glucopyranosyl side-branching unit	[76]
PUP-W-1	*Polyporus umbellatus sclerotia*	41.1 kDa	Only glucose	The backbone consisted of a repeating chain of eight →3)-β-D-Glcp-(1→ units, with branched chains of four β-D-Glcp residues, joined by repeating 1,6-linkage units at the O-6 position of the backbone	[77]
PGPS	*Polyporus grammocephalus*	140 kDa	Only glucose	The repeating unit of the glucan was →3)-α-D-Glcp(1→[4) -α-D-Glcp(1]_2_→	[78]
OL-2	*Omphalia lapidescens*	N/A	Only glucose	It consisted of a 1,3-β-Glucan backbone chain decorated with a single six-branched β-glucosyl side unit on every fourth residue	[79]
TFP-1	*Lasiosphaera fenzlii* fruit body	500 kDa	Glucose:Mannose:Galactose = 6.3:1.03:2.07	Total sugar content of 96.94%; Terminal Glcp:1,3-linked Galp:1,2-linked Manp:1,3-linked Glcp:1,6-linked Glcp:1,2,3,6-linked Glcp = 28.48:10.31:26.69:25.86:8.21:8.25	[80]
TFP-2	*Lasiosphaera fenzlii* fruit body	500 kDa	Glucose:Mannose:Galactose = 8.68:0.69:0.63	Total sugar content of 97.24%; Terminal Glcp:1,3-linked Galp:1,2-linked Manp:1,3-linked Glcp:1,6-linked Glcp:1,2,3,6-linked Glcp = 14.42:6.27:9.71:30.85:34.03:8.75	[80]
TFP-3	*Lasiosphaera fenzlii* fruit body	600 kDa	Glucose:Mannose:Galactose = 7.46:1.25:1.29	Total sugar content of 95.24%; 1,3-linked Galp:1,2-linked Manp:Terminal Glcp:1,3,6-linked Glcp:1,2,3,6-linked Glcp = 0.132:0.12:0.402:0.261:0.075	[80]
TFP-4	*Lasiosphaera fenzlii* fruit body	1000 kDa	Glucose:Mannose:Galactose = 5.89:1.22:2.89	Total sugar content of 56.94%; uronic acid content of 47.95%; protein content of 34.89%; due to the high content of protein and uronic acid, methylation was not successful	[80]
CGP I-1	*Calvatia geigantea*	N/A	Glucose:Mannose:Galactose = 11.28:1.22:3.92	The CGP I-1 molecules was a linear molecules with branches; the main chain is composed of mannose and glucose, and there were 1→3, 1→2,3, 1→3,6 bond types that were not oxidized by periodate acid; the branch chain or end residues of the main chain were composed of β-Gal(1→4), β-Glc(1→6) and α-Glc(1→4)	[81]

## Data Availability

Not applicable.

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
