# Peer review of "A Systematic Review on the Research Progress on Polysaccharides from Fungal Traditional Chinese Medicine"

_molecules, 2023, doi:10.3390/molecules28196816_

Round 1

Reviewer 1 Report

The topic of this Review is interesting and can be considered for publication in Molecules after some revisions. 

There are several issues regarding the use of italics when scientific names are written, the format for citing references in the text should include only the last name of the first author followed by et al. 

 Other specific comments are listed below

L31, 49-50, etc. Ensure that all the scientific names are in italics.

Table 1. Use italics for scientific names, and homogenize the format for the volume units, L, and mL. Instead of using “ml”. Use “0.15 M NaOH solution” instead of “0.15 mol NaOH solution”. Homogenize the number of significant figures in the Total yield column.

L87 & others. Add the period at the end of et al.

L89. Procure not repeat information already given in the Table.

L95-96. Revise the wording for this phrase “All figures and tables should be cited in the main text as Figure 1, Table 1, etc.”

L101-102. Revise the wording for this sentence, it is not clear.

L159. It is suggested to use “extraction yield” instead of “extraction rate”. The reported units do not agree with those associated with a rate.

Table 2. Homogenize the use of NaCl or sodium chloride.

L226-227. Homogenize the format for citing references based on the guide to authors' statements.

Section 2.2. It is suggested to consider grouping the studies based on the type of fungi raw material, cultures, mycelia, fruit body, etc., perhaps following a similar grouping as in Section 3. Also, consider adding information about the type of chromatography used for each fractionation column, i.e., gel permeation, ionic interaction, hydrophobic interaction, etc., when describing the procedures applied during the purification of polysaccharides in the different studies. Not all the commercial resins described in this section include the type of chromatography. As mentioned above, revise the wording for scientific names, and put them in italics.

L247. Indicate the composition of the eutectic ionic solvents.

L249. Include more quantitative evidence to support the statement about “strong antioxidant activity”.

L249-257. The wording in this paragraph should be revised and improved in the description of the purification processes involved in each work cited.

L262-263. Ensure that all the descriptions of purification processes reported in this section are given and appropriately described in the text.

L325. Ensure this nomenclature is defined or clearly referred to as the name of the polysaccharide given in the work cited in the text. Revise the document.

L336. It seems that a closed parenthesis is missing.

L350. Use “higher than 90%” instead of “more than 90%”.

L361-363. Include a citation for this statement.

Section 3.5. Is there any information about the chemical composition or any characteristic of this glucan material?

L403. 500 kDa is repeated.

Table 3. Homogenize the number of significant figures (one or two digits), and the units (kDa or g/mol).

L414-415. Put in italics terms like in vivo and in vitro.

Figure 2. Ensure the correct contrast between the color of the font and the background.

Table 4. Homogenize the units for mL, instead of ml.

Some citations in the text do not fulfill the format required for citing in the text, the name of the author should be deleted. Scientific names in the titles of the references listed should be in italics. 

L876. Check the spelling for this reference. Li instead of LI.

Some sections require improvement for clarifying some statements or improving the information's organization.

Reviewer 2 Report

The review

A systematical review on the research progress of polysaccharides from fungal traditional Chinese medicine" is novel and as per the scope of journal.

The review Indepth and very well compiled. The authors touched all most all part which are necessary for readers. They explained and summarized extraction, purification methods, structure and condition parameters of polysaccharides from fungal traditional Chinese medicine. In table 4 they also summited the bioactivities and mechanisms of polysaccharides of this medicine.

Comments:

1. Stability and regulatory status of this medicines.

2. Summarization of Clinical studies performed on these medicines.

Round 2

Reviewer 1 Report

The authors have attended all the comments, the manuscript can be accepted in its current state,